# RNF208, an estrogen-inducible E3 ligase, targets soluble Vimentin to suppress metastasis in triple-negative breast cancers

Kyoungwha Pang [1,2], Jinah Park[1], Sung Gwe Ahn[3], Jihee Lee[1,2], Yuna Park[1,2], Akira Ooshima[1], Seiya Mizuno[4], Satoshi Yamashita[5], Kyung-Soon Park[2], So-Young Lee[6], Joon Jeong[3], Toshikazu Ushijima [5], Kyung-Min Yang[1]* & Seong-Jin Kim[1,7,8]*

The development of triple-negative breast cancer (TNBC) negatively impacts both quality of life and survival in a high percentage of patients. Here, we show that RING finger protein 208 (RNF208) decreases the stability of soluble Vimentin protein through a polyubiquitin-mediated proteasomal degradation pathway, thereby suppressing metastasis of TNBC cells. RNF208 was significantly lower in TNBC than the luminal type, and low expression of RNF208 was strongly associated with poor clinical outcomes. Furthermore, RNF208 was induced by 17β-estradiol (E2) treatment in an estrogen receptor alpha (ERα)-dependent manner. Overexpression of RNF208 suppresses tumor formation and lung metastasis of TNBC cells. Mechanistically, RNF208 specifically polyubiquitinated the Lys97 residue within the head domain of Vimentin through interaction with the Ser39 residue of phosphorylated Vimentin, which exists as a soluble form, eventually facilitating proteasomal degradation of Vimentin. Collectively, our findings define RNF208 as a negative regulator of soluble Vimentin and a prognostic biomarker for TNBC cells.

[1] Precision Medicine Research Center, Advanced Institute of Convergence Technology, Seoul National University, Suwon, Gyeonggi-do 16229, Republic of Korea. [2] Department of Biomedical Science, College of Life Science, CHA University, Seongnam City, Gyeonggi-do 463-400, Republic of Korea. [3] Department of Surgery, Gangnam Severance Hospital, Yonsei University Medical College, 712 Eonjuro, Gangnam-Gu, Seoul 135-720, Republic of Korea. [4] Laboratory Animal Resource Center, University of Tsukuba, Tsukuba, Ibaraki 305-8575, Japan. [5] Division of Epigenomics, National Cancer Center Research Institute, Tokyo, Japan. [6] Department of Internal Medicine, CHA University, Seongnam City, Gyeonggi-do 463-400, Republic of Korea. [7] Department of Transdisciplinary Studies, Graduate School of Convergence Science and Technology, Seoul National University, Suwon, Gyeonggi-do 16229, Republic of Korea. [8] TheragenEtex Bio Institute, TheragenEtex Co, Suwon, Gyeonggi-do 16229, Republic of Korea. *email: yangkm@snu.ac.kr; jasonsjkim@snu.ac.kr

Breast cancer is one of the most common cancers worldwide among women[1]. Among the molecular subtypes of breast cancer, ERα-positive breast tumors are a well-differentiated phenotype and are correlated with a better prognosis than ERα-negative breast tumors, which are extremely aggressive subtypes associated with a poor prognosis[2,3]. Although ERα-positive breast tumors initially respond to hormonal manipulation, such as tamoxifen, which is an antiestrogen agent, patients with triple-negative breast cancer (TNBC) or ERα-negative breast tumors do not benefit from antihormonal therapy or traditional chemotherapy[4–6]. In this regard, different prognostic and therapeutic applications based on molecular features are necessary for breast cancer patients.

Dysregulation of ERα is often associated with advanced breast cancers by altering the ERα-dependent genes that are involved in cell proliferation and metastasis. The loss of *ESR1* expression in TNBC is a result of the hypermethylation of specific CpG islands within the *ESR1* promoter through regulation of DNA methyltransferase (DNMT)[7,8]. Furthermore, ERα re-expression by 5-aza-dC, a DNMT inhibitor, was shown to inhibit tumor growth of TNBC cells in vitro and in vivo, indicating that its expression can be modulated by epigenetic mechanisms and restore the sensitivity of TNBC to endocrine therapy[9]. Although the function of ERα has been extensively studied in breast cancer, how the loss of ERα contributes to tumorigenesis and metastasis in TNBC is unclear.

Vimentin, an intermediate filament protein, is highly expressed in aggressive epithelial cancers, including breast cancer, prostate cancer, gastric cancer, malignant melanoma, and lung cancer, where its expression level is associated with increased risks of metastasis[10]. Aberrant expression of Vimentin is restricted to TNBC cells among the breast cancer cells and is involved in a mesenchymal phenotype, aggravating the invasive potential of breast cancer cells. Vimentin regulates cell adhesion and motility through its phosphorylation (soluble form) and dephosphorylation (insoluble form)[11–13]. Furthermore, posttranslational modifications (PTMs), such as O-linked glycosylation, ubiquitination, sumoylation, and acetylation, are known to regulate the function of Vimentin[14]. Recently, the E3 ubiquitin ligase TRIM56 was revealed to be a negative regulator of Vimentin by inducing polyubiquitination-mediated proteasomal degradation, resulting in a decrease of cell migration and invasion[15]. Although Vimentin function is widely studied in cancer metastasis, the molecular mechanisms by which the suppression of Vimentin ameliorates metastasis in aggressive cancer cells remain to be identified.

In this study, we show RNF208, an estrogen-inducible E3 ligase protein, specifically induces degradation of soluble Vimentin through K27-linked polyubiquitination of phosphorylated Vimentin at the Ser39 residue, thereby suppressing the metastasis of aggressive TNBC cells.

## Results

**RNF208 is significantly underexpressed in aggressive TNBC.** To identify biomarkers associated with breast cancer progression, we initially performed RNA sequencing in several breast cancer cell lines, classified as either the luminal subtypes (MCF-7, T47D, ZR-75B) or TNBC (MDA-MB-231, Hs578T). Based on the transcriptome analysis, we found that *RNF208* was significantly underexpressed in TNBC cells compared to luminal breast cancer cells (GSE100878) (Fig. 1a). This finding was supported by RT-PCR and immunoblot analysis (Fig. 1b). Moreover, gene expression analysis of a public microarray dataset (GSE41313) with 52 breast cancer cell lines showed significantly lower expression of *RNF208* in TNBC cells compared to luminal cells

(Fig. 1c). We further analyzed *RNF208* expression in different breast cancer subtypes using microarray and RNA sequencing datasets of breast cancer patients obtained from Genomic Data Commons (GDC) datasets and public microarray datasets (GSE2034). Consistently, *RNF208* expression was significantly decreased in patients with TNBC compared to luminal A, luminal B, and HER2-enriched patients (Fig. 1d−f). We next examined the RNF208 protein and mRNA levels in human primary breast tumor specimens by immunohistochemistry staining and quantitative RT-PCR. Notably, the expression of RNF208 was remarkably lower in the tumor compartments of patients with TNBC compared to those with the luminal subtypes (Fig. 1g), and *RNF208* mRNA was significantly decreased in TNBC tissues (Fig. 1h). To address the functional significance of the association of RNF208 expression with clinical outcomes in breast cancers, we performed public meta-analyses using Kaplan−Meier Plotter software[16]. Interestingly, patients with low *RNF208* expression exhibited significantly shorter relapse-free survival times than those with high expression (Fig. 1i). Moreover, the low expression of *RNF208* was strongly correlated with poor relapse-free survival, regardless of breast cancer subtypes (Supplementary Fig. 1). Taken together, these results suggest that RNF208 is closely associated with aggressive breast cancer and may serve as a predictive factor for the risk of developing relapse disease in breast cancer patients.

**ERα regulates RNF208 expression.** Because RNF208 expression was increased in luminal breast cancer subtypes, which are ERα-positive, we investigated the relationship between RNF208 and ERα expression using public microarray datasets (GSE2034; GSE5460). The expression of RNF208 was positively correlated with the ERα status in breast cancers (Fig. 2a). Next, we tested whether RNF208 may be a bona fide estrogen-responsive target gene. MCF-7 and T47D cells were cultured in phenol-free media with charcoal stripped serum to remove residual exogenous estrogens and estrogen-like compounds. Interestingly, RNF208 expression was markedly induced by E2 treatment, similar to other ERα-responsive genes, such as *FOXOM1* and *GREB1* (Fig. 2b), and siRNA-induced *ESR1* knockdown attenuated the E2-induced expression level of *RNF208* in T47D cells (Fig. 2c), indicating that RNF208 expression may depend on ERα expression in luminal breast cancer cells. To determine whether the expression of RNF208 induced by E2 is associated with direct transcriptional activation of the RNF208 promoter, we generated deletion constructs of the RNF208 promoter, which contains estrogen-responsive elements (EREs), inserted upstream of a pGL3 luciferase reporter gene (Fig. 2d; Supplementary Fig. 2a). Notably, E2 treatment increased the luciferase activity of the RNF208 promoter fragment up to −1154 bp, whereas deletion of position −1121 bp or −724 bp was not responsive to E2 treatment in T47D and MCF-7 (Fig. 2e; Supplementary Fig. 3a). To determine whether E2-induced activation of the RNF208 promoter increased the direct binding of ERα to the RNF208 gene locus, we conducted chromatin immunoprecipitation (ChIP) using the anti-ERα antibody in T47D cells. E2 treatment dramatically increased the interaction of ERα within the −1131 bp to −1124 bp promoter region, which contains one ERE site, indicating that endogenous ERα is recruited to the RNF208 promoter (Fig. 2f). To further characterize which ERE site located in the −1131 bp to −1124 bp region is responsible for E2-induced activation of the RNF208 promoter, we generated an ERE mutant construct (mtR4) carrying two substitutions (GACC to GAAA). The luciferase activity of the full-length RNF208 promoter was increased by E2 treatment in T47D cells and MCF-7, whereas the ERE mutation markedly abolished RNF208 promoter activation

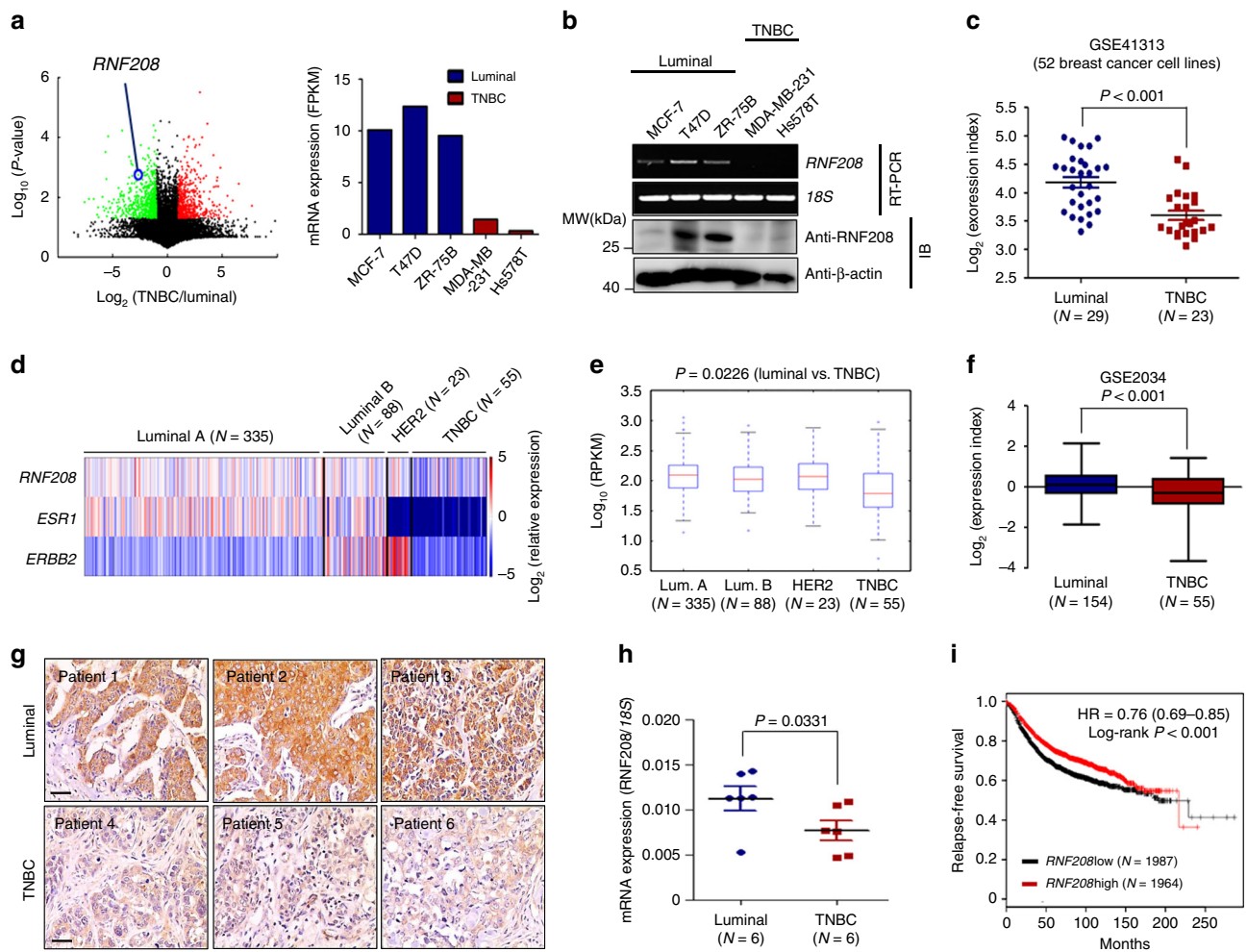

**Fig. 1 RNF208 is significantly underexpressed in highly metastatic breast cancer cells. a** Volcano plot of *RNF208* expression in luminal subtype (MCF-7, T47D, and ZR-75B) and TNBC (MDA-MB-231 and Hs578T) cells using RNA sequencing (GSE100878). **b** RT-PCR and immunoblot analysis of RNF208 expression in breast cancer cells. **c** Scatter plots of *RNF208* expression in 52 breast cancer cells from public microarray datasets (GSE41313) (Luminal $N =$ 29 and TNBC $N =$ 23 cells). **d, e** Heat map and box plots showing the expression levels of *RNF208, ESR1*, and *ERBB2* across the breast cancer subtypes using Genomic Data Commons (GDC). (Luminal A $N =$ 335, Luminal B $N =$ 88, HER2 $N =$ 23 and TNBC $N =$ 55 patients). **f** Comparison of *RNF208* expression in breast cancer subtypes using published microarray datasets (GSE2034) (Luminal $N =$ 154 and TNBC $N =$ 55 patients). **g** Representative immunohistochemical (IHC) staining of RNF208 protein expression classified in luminal and basal breast cancer patient tissues. Original magnification ×100. Scale bar, 50 μm. **h** Real-time quantitative RT-PCR of *RNF208* expression in luminal subtype tissues ($N =$ 6) and TNBC tissues ($N =$ 6); 18S rRNA was used as an internal control. **i** Kaplan−Meier analysis showing relapse-free survival depending on RNF208 expression levels from public meta-analysis data ($N =$ 3951). $P$ values were calculated using a log-rank test. All $P$ values were calculated by unpaired two-tailed Student's $t$ tests (**c**, **e**, **f**, and **h**). These data represent the mean ± S.D. Source data for (**c**, **f**, **h**) is available in Source Data file. Unprocessed original scans of blots and gels in (**b**) are shown in Supplementary Fig. 13.

(Fig. 2g; Supplementary Fig. 3b). Furthermore, deletion of −1131/−1124 bp region of RNF208 promoter markedly attenuated the E2-induced RNF208 expression in MCF-7 cells (Fig. 2h). As RNF208 is poorly expressed in TNBC cells, we next investigated whether downregulation of RNF208 is a result of aberrant DNA methylation within its promoter in TNBC cells using a public genome database (GSE68379). The methylation levels of the RNF208 promoter were not significant in TNBC cells, indicating that methylation is not involved in RNF208 silencing (Supplementary Fig. 2). Considering that *ESR1* expression is suppressed by epigenetic mechanisms, such as DNA methylation, in TNBC cells, we further examined whether low expression of RNF208 in TNBC cells might be due to silencing of *ESR1* expression by DNA methylation. Interestingly, 5-aza-dC treatment resulted in an increase in the expression levels of ERα and RNF208 in TNBC cells (MDA-MB-231, Hs578T) (Fig. 2i, Supplementary Fig. 3c). More strikingly, ERα knockdown by siRNA attenuated the 5-aza-

dC-induced increase of RNF208 in TNBC cells, implying that ERα expression is required for the induction of RNF208 (Fig. 2j). Taken together, our results suggest that RNF208 is a direct estrogen-responsive target gene that is dependent on ERα.

**RNF208 overexpression reduces the tumor growth and lung metastasis.** Our findings led us to verify the physiological function of RNF208 in breast cancer progression. We first investigated whether the expression level of RNF208 affects tumorigenesis in vitro and in vivo. RNF208 overexpression markedly reduced the proliferation of TNBC cells (Fig. 3a). To further test the effect of RNF208 on the tumorigenic capacity of TNBC cells in vivo, we injected RNF208-tolu TNBC cells subcutaneously into the flanks of immunodeficient mice. Overexpression of RNF208 dramatically attenuated the tumor volume of TNBC cells (Fig. 3b). Furthermore, expression of Ki-67, a marker of cell proliferation,

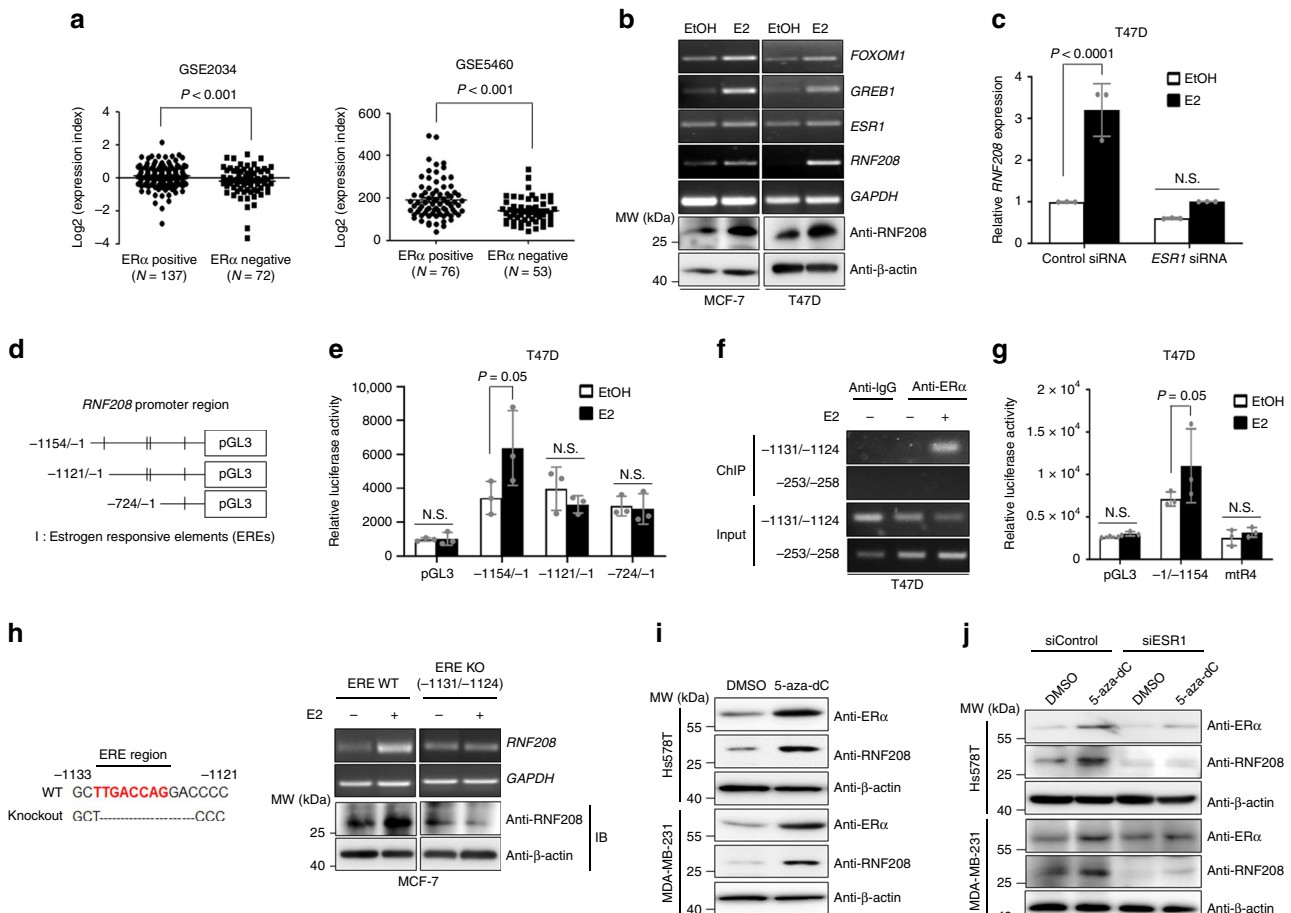

**Fig. 2 RNF208 expression was transcriptionally activated by ERα in luminal breast cancer cells. a** Comparison of *RNF208* expression between ERα-positive and -negative breast cancer samples in published microarray datasets (GSE2034 and GSE5460). **b** RT-PCR (top) and immunoblot (bottom) analysis of ERα target genes (*FOXOM1, GREB1, ESR1*) and RNF208 expression in EtOH- or E2-treated ERα-positive breast cancer cells. 10 nM or 20 nM of E2 was treated in MCF-7 or T47D cell, respectively and E2 was treated for 24 h for RT-PCR and 48 h for immunoblot analysis. *GAPDH* and β-actin were used as internal controls. **c** Real-time quantitative RT-PCR (qRT-PCR) of *RNF208* expression in *ESR1*-knockdown T47D cells upon E2 treatment. **d** Illustration of luciferase reporters including ERα-binding sites in the RNF208 promoter sequences. **e** T47D cells were transfected with various deletion constructs of the RNF208 promoter and then treated with or without E2 for 24 h. After E2 treatment, cells were assayed for luciferase activity. **f** ChIP analysis showing the recruitment of ERα to the human RNF208 promoter in E2-treated T47D cells. Precipitation was conducted with antinormal IgG or anti-ERα antibodies. **g** T47D cells were transfected with pGL3 control, RNF208 promoter, or its mutant (mtR4) plasmids containing mutated estrogen-responsive element site (CACC sequence replaced by GAAA) and then subjected to luciferase assays. **h** RT-PCR and immunoblot analysis showing RNF208 expression upon E2 treatment for 24 h (mRNA) or 48 h (protein), respectively, in control or ERE knockout MCF-7 cells. **i** Immunoblot analysis of ERα with 5-aza-dC treatment in TNBC cells. Hs578T and MDA-MB-231 cells were treated with 10 μM 5-aza-dC for 96 h and β-actin was used for normalization. **j** TNBC cells were transiently transfected with 20 nM of control siRNA or *ESR1* siRNA and then treated with 10 μM 5-aza-dC for 96 h. Cell lysates were immunoblotted with the indicated antibodies. All *P* values were calculated by unpaired two-tailed Student's *t* tests (**e, g**). These data represent the mean ± SD of three independent experiments. Source data for (**c, e, g, h**) are available in Source Data file. Unprocessed original scans of blots and gels in (**b, f, h–j**) are shown in Supplementary Fig. 13.

was decreased (Fig. 3c; Supplementary Fig. 4a), whereas active caspase-3, a marker of cell death, was increased in RNF208-overexpressing primary tumor tissues compared with control tissues (Fig. 3d; Supplementary Fig. 4b).

Because tumorigenic potential is closely associated with enhanced invasiveness for metastasis of tumor cells, we next examined whether RNF208 affects cell migration and invasiveness using Transwell migration and Matrigel invasion assays. Overexpression of RNF208 significantly decreased cell migration and invasion in Hs578T and MDA-MB-231 cells (Fig. 3e, f). We further observed that RNF208-overexpressing MDA-MB-231 cells showed a significant reduction in lung metastasis as well as metastatic pulmonary nodules in immunodeficient mice (Fig. 3g, h). We then investigated whether the loss of RNF208 increases the aggressive potential in the luminal subtype of breast cancer cells. Unexpectedly, neither knockdown nor knockout of RNF208 did change the expression of epithelial to mesenchymal transition (EMT) markers, such as E-cadherin and Vimentin, or the cell migration ability of both control and RNF208-deficient T47D cells (Supplementary Fig. 5), suggesting that loss of RNF208 may not be sufficient to induce aggressive phenotypes and cancer progression in luminal breast cancer subtypes. Taken together, these results strongly suggest that RNF208 may act as a tumor suppressor in the cancer progression of TNBC cells.

**RNF208 overexpression decreases the stability of Vimentin protein.** To assess the molecular mechanism underlying RNF208 function in breast cancer progression, we performed a mass spectrometry-based formaldehyde crosslinking assay in RNF208-overexpressing MDA-MB-231 cells. Through mass spectrometry

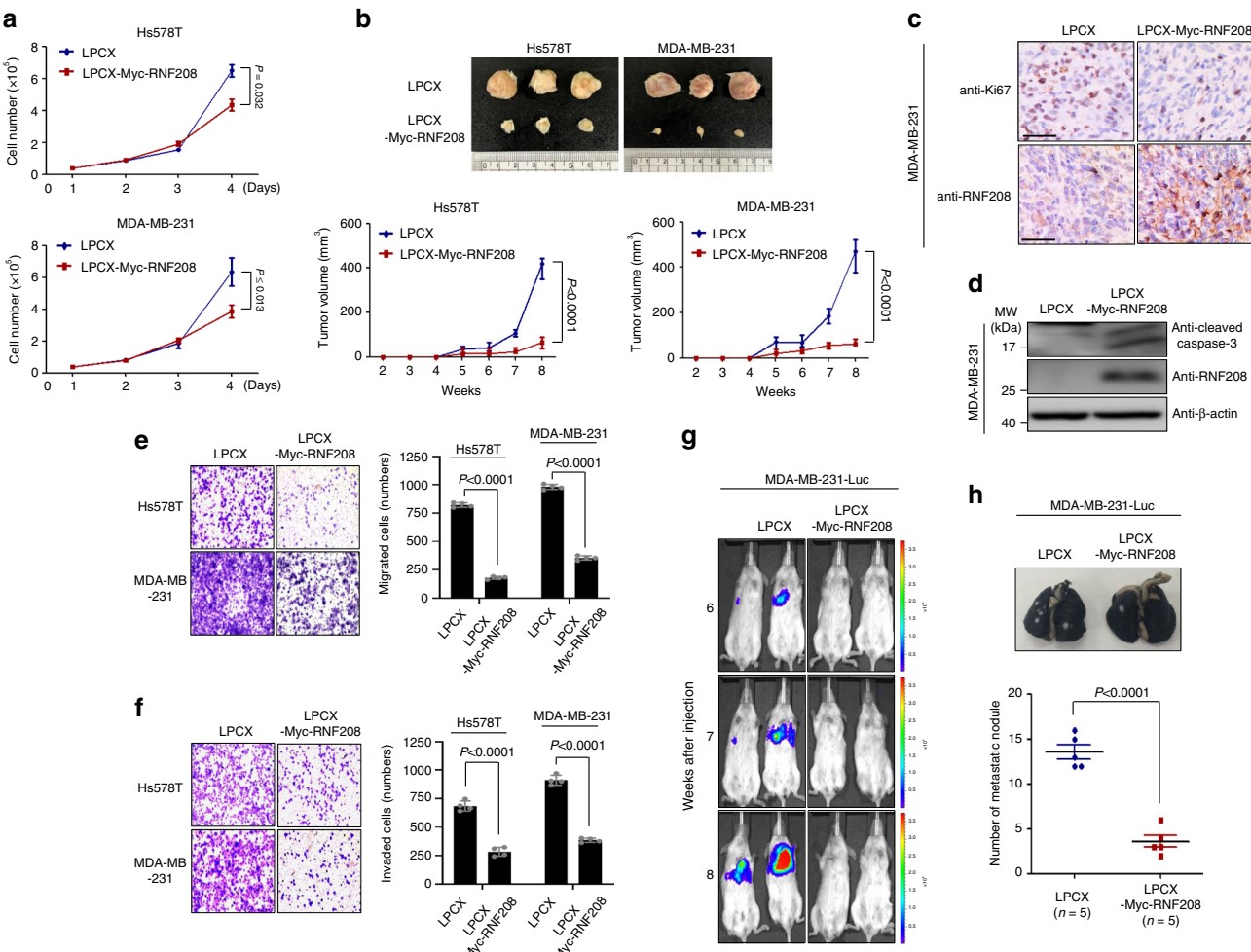

**Fig. 3 Overexpression of RNF208 reduces tumorigenesis and lung metastasis in TNBC cells. a** Cell doublings of RNF208-overexpressing Hs578T and MDA-MB-231 cells. Each point represents the mean of cell numbers counted in triplicate dishes. **b** Tumor formation and growth of control and RNF208-overexpressing TNBC cells subcutaneously injected into the flanks of NOD/SCID mice ($n = 6$ per group). Primary tumor volumes were measured weekly, and mice were sacrificed at 8 weeks (Hs578T and MDA-MB-231). Representative primary tumor images (top) and tumor volumes (bottom) are shown. **c** Representative IHC image showing Ki-67 and RNF208 expression in primary tumor tissues from (**b**). Original magnification ×100. Scale bar, 100 μm. **d** Immunoblot analysis of active caspase-3 expression using lysates of RNF208-overexpressing MDA-MB-231 cells. β-actin was used as an internal control. **e, f** Transwell migration and Matrigel invasion assays of TNBC cells stably expressing RNF208 proteins. After 24 h, migrated (**e**) and invaded (**f**) cells were counted following staining with crystal violet. **g** Representative bioluminescent (BLI) imaging of NOD/SCID mice ($n = 5$ per group) showing lung metastasis from 6 to 8 weeks, derived from lateral tail-vein injection of control and RNF208-overexpressing MDA-MB-231-Luc. **h** Representative whole lung image stained with India ink showing metastatic nodules (top) and scatter plot showing the number of lung metastatic nodules (bottom). All P values were calculated by unpaired two-tailed Student's t tests (**a, b,** and **h**). The data represent the mean ± SD. Source data for (**a, b, e, f, h**) are available in Source Data file. Unprocessed original scans of blots and gels in (**d**) is shown in Supplementary Fig. 13.

analysis, Vimentin was identified as a potential binding protein of RNF208 (Fig. 4a). Because Vimentin is associated with the mesenchymal phenotype of aggressive cancer cells in metastatic cancer progression, we first investigated whether RNF208 regulates the expression of EMT markers in Hs578T and MDA-MB-231 cells. Interestingly, RNF208 overexpression strongly decreased the expression level of Vimentin protein without affecting its mRNA or E-cadherin expression (Fig. 4b, c). To determine whether the reduced expression level of Vimentin in RNF208-overexpressing cells is an account of its attenuated protein stability, we measured the half-life of Vimentin protein after treatment with cycloheximide, a protein synthesis inhibitor. RNF208-ovexpressing MDA-MB-231 cells showed a faster rate of Vimentin degradation compared to control cells (Fig. 4d). Moreover, decreased Vimentin stability under RNF208 overexpression was rescued by treatment with the proteasome

inhibitor MG132 but not by autophagy inhibitors such as bafilomycin A1 (BAF), chloroquine (CQ), and 3-methyladenine (3-MA) (Supplementary Fig. 6b). Consistent with this observation, decreased stability of Vimentin in cells with ectopic expression of RNF208 was also markedly restored by MG132 treatment (Fig. 4e).

Since Vimentin is identified as a binding partner of RNF208, we next tested the interaction between RNF208 and Vimentin by coimmunoprecipitation in 293T and Hs578T cells. As expected, immunoprecipitation assays showed either exogenous or endogenous interactions between these two proteins (Fig. 4f; Supplementary Fig. 6c). Considering the ubiquitin transfer activity of RNF208 E3 ligase, we next investigated whether RNF208 mediates the ubiquitination of Vimentin. Notably, ectopic overexpression of RNF208 induced the polyubiquitination of Vimentin (Fig. 4g). To pinpoint whether RNF208-polyubiquitinated Vimentin is linked to

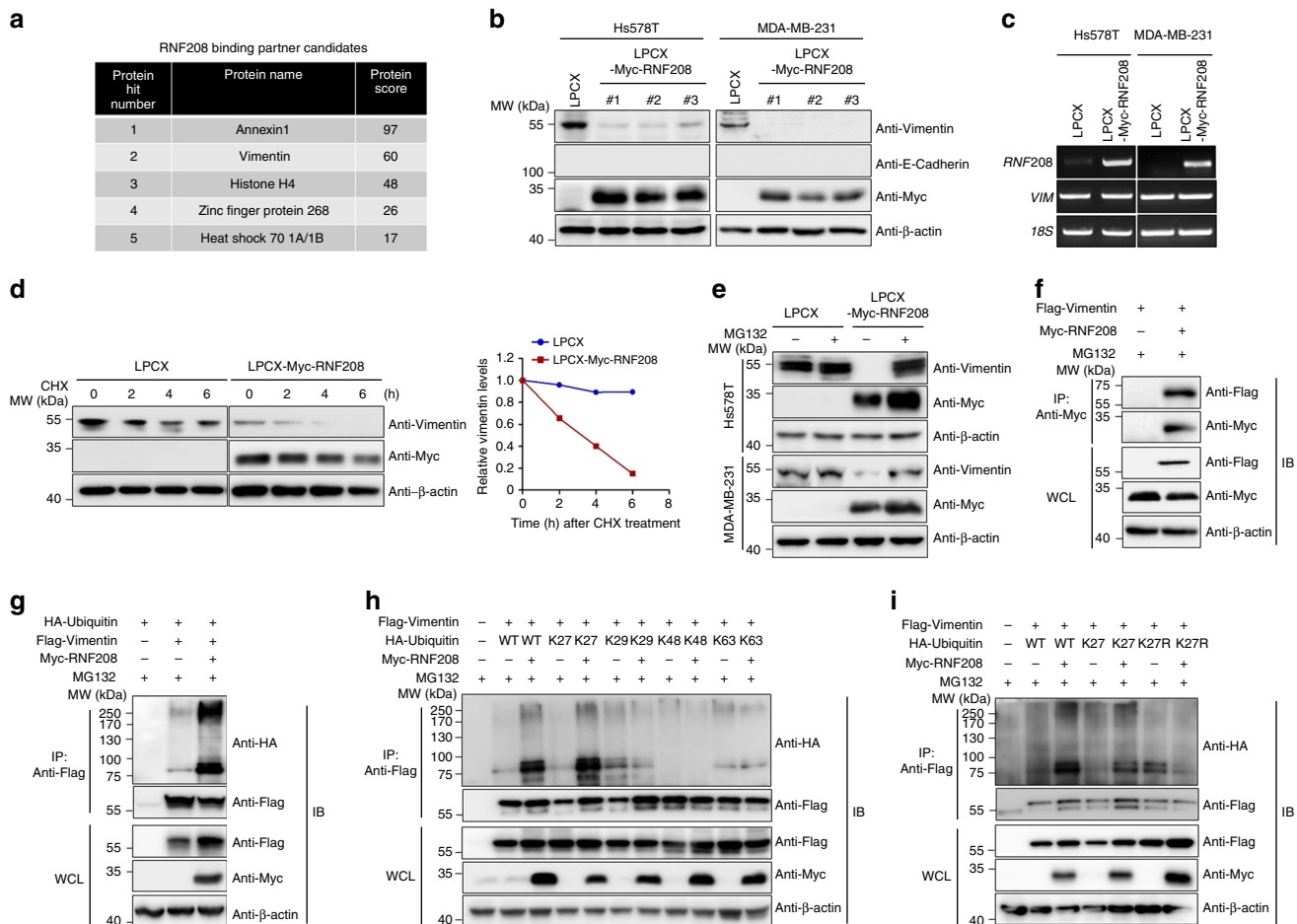

**Fig. 4 RNF208 induces the proteasomal degradation of Vimentin protein by facilitating K27-linked polyubiquitination. a** List of RNF208 binding partner candidates analyzed from the mass spectrometry-based crosslinking assay. **b** Immunoblot analysis showing Vimentin, E-cadherin, and RNF208 in control and RNF208-overexpressing Hs578T and MDA-MB-231 clonal cells (#1−#3). β-actin was used as an internal control. **c** RT-PCR showing *RNF208* and *VIM* expression in control and RNF208-overexpressing Hs578T or MDA-MB-231 cells; *18S* was used as an internal control. **d** Immunoblot analysis showing the stability of the Vimentin protein in control and RNF208-overexpressing MDA-MB-231 cells in the presence of cycloheximide (CHX, 50 μg ml⁻¹) for the indicated times (left). The data were quantified using ImageJ software (right), and β-actin expression was used for normalization. **e** Immunoblot analysis showing the expression level of Vimentin protein in control and RNF208-overexpressing Hs578T and MDA-MB-231 cells with the proteasome inhibitor MG132. Cells were treated with 10 μM MG132 for 6 h, and cell lysates were subjected to immunoblotting with the indicated antibodies. **f** Immunoprecipitation assay showing the endogenous interaction between RNF208 and Vimentin. Cells were cotransfected with Myc-RNF208 and Flag-Vimentin plasmids in 293T cells upon MG132 treatment. Cell lysates were immunoprecipitated with anti-Myc antibody and then immunoblotted with the indicated antibodies. WCL whole-cell lysates. **g** Immunoprecipitation assay showing the ubiquitination of Vimentin by ectopic expression of RNF208. 293T cells were cotransfected with HA-Ubiquitin, Flag-Vimentin, or Myc-RNF208 plasmids in 293T cells upon MG132 treatment. Vimentin ubiquitination was detected by immunoprecipitation with anti-Flag and then immunoblotted with the indicated antibodies. **h** Immunoprecipitation assay showing K27-mediated ubiquitination of Vimentin by RNF208. The 293T cells were cotransfected with Flag-Vimentin, different linkages of HA-Ubiquitin (wild-type, K27, K29, K49. K63), or Myc-RNF208 plasmids, and then, cell lysates were immunoprecipitated with anti-Flag antibody. **i** Flag-Vimentin plasmid was cotransfected into 293T cells together with wild-type, K27 or lysine mutant (K27R) of HA-Ubiquitin in the absence or presence of Myc-RNF208 plasmid. Cells lysates were immunoprecipitated with anti-Flag antibody and then immunoblotted with the indicated antibodies. Unprocessed original scans of blots in (**b**−**i**) are shown in Supplementary Fig. 13.

any type of polyubiquitin chains, 293T cells were transiently transfected with Flag-Vimentin together with HA-Ubiquitin (wild-type, K27, K29K, K48, and K63) in the presence or absence of Myc-RNF208 upon MG132 treatment. Interestingly, RNF208 specifically promoted a robust polyubiquitination of Vimentin by K27-only ubiquitin (HA-K27 Ub) in the same pattern as wild-type HA-Ub, not by any other chain types (Fig. 4h). Further experiments using the K27R ubiquitin mutant, in which the lysine 27 residue of ubiquitin was mutated to arginine (HA-K27R Ub), revealed that RNF208-induced polyubiquitination of Vimentin was not detected in the presence of K27R, indicating that the polyubiquitination of Vimentin by RNF208 may be dependent on K27 ubiquitin-linked chain formation (Fig. 4i). An immunoprecipitation assay showed

that RNF208 overexpression induced an endogenous K27-linked polyubiquitination of Vimentin in MDA-MB-231 cells (Supplementary Fig. 6c). Taken together, our results suggest that RNF208 decreases the stability of Vimentin through K27 ubiquitin-linked polyubiquitination, positioning RNF208 as a novel negative regulator of Vimentin in TNBC cells.

**RNF208 expression is inversely correlated with Vimentin expression.** We next prompted an investigation of the relevance between RNF208 and Vimentin expression in patients with breast cancer. To this end, we used breast cancer TMAs obtained from the Seoul National University College of Medicine in South

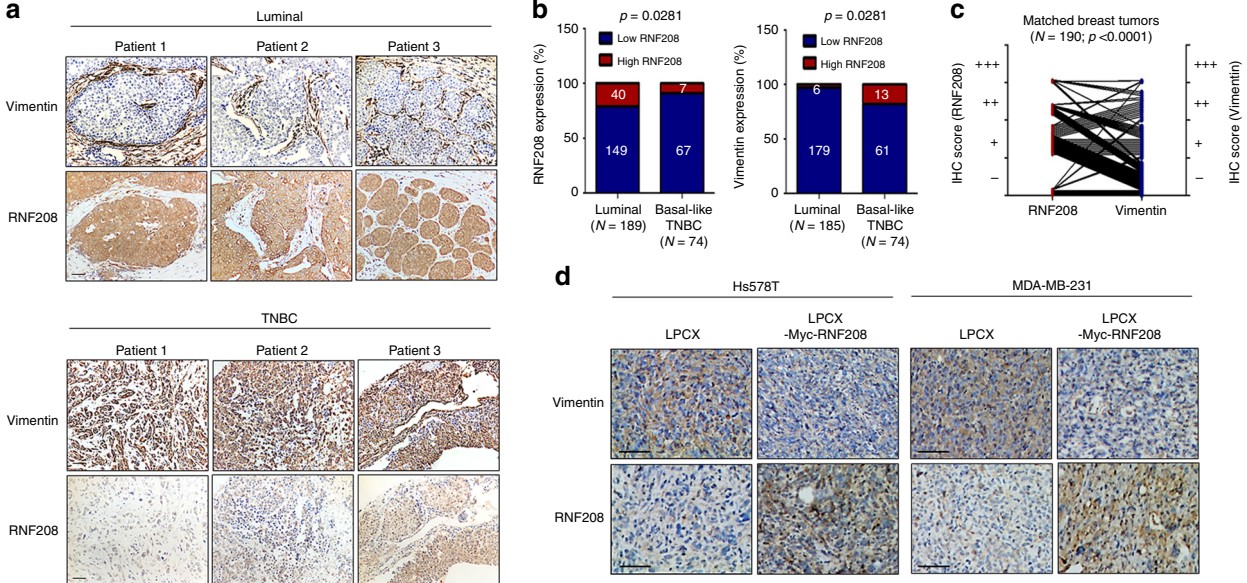

**Fig. 5 RNF208 expression is inversely correlated with Vimentin expression in patients with breast cancer. a** Representative IHC images showing the expression of RNF208 and Vimentin in luminal subtype or TNBC patient tumor tissues. Original magnification, ×100. Scale bar, 50 μm. **b** The stacked bar charts showing the expression of RNF208 and Vimentin from TMAs of breast cancer patients (Luminal for RNF208 ($N = 189$), Vimentin ($N = 185$) and basal-like TNBC $N = 74$). $P$ values were calculated by Fisher exact test. Numbers in the bar charts indicate the number of patients according to IHC intensity scores. **c** The graph showing the RNF208 and Vimentin IHC intensity in matched tumors from TMAs of breast cancer patients ($N = 190$). $P$ values were calculated by paired two-tailed Student's $t$ tests. **d** Representative IHC images showing RNF208 and Vimentin expression in primary tumor tissues from Fig. 3b. Original magnification, ×100. Scale bar, 50 μm. Source data for (**b**, **c**) are available in Source Data file.

Korea. Notably, RNF208 and Vimentin were inversely expressed in patients with luminal subtype cancer and TNBC tissues (Fig. 5a, b). To further support this observation, we examined matched tumor samples in patients with breast cancer. Indeed, a significant inverse relationship between RNF208 and Vimentin expression was observed in matched tumor tissues of patients with breast cancer (Fig. 5c). Furthermore, expression of Vimentin was decreased in RNF208-ovexexpressing primary tumor tissues compared with control tissues in xenograft mouse experiment (Fig. 5d). Taken together, these results suggest that RNF208 expression is closely associated with Vimentin-mediated aggressive cancer progression of breast cancer cells.

**Activity of RNF208 is required for the degradation of Vimentin.** Considering that RNF208 contains a RING domain, which possesses E3 ligase activity for ubiquitination, we assumed that the RING domain of RNF208 might interact with Vimentin to facilitate polyubiquitination. Indeed, an immunoprecipitation assay showed that Vimentin specifically bound to the RING domain of RNF208, which harbors the E3 ligase activity, but not to the N-terminal and C-terminal regions (Supplementary Fig. 7a). In parallel with this result, given that the cysteine-rich motifs are known as the active site of E3 ligase activity, we also examined whether activity of RNF208 is responsible for the polyubiquitination of Vimentin. To this end, we generated RNF208 mutants (C143A, C167A, and C186A) in which putative cysteine-rich motifs in the RING domain of RNF208, which are highly conserved in different species (Supplementary Fig. 7b), were substituted with alanine. Importantly, the inactive mutants of RNF208 did not bind to Vimentin (Fig. 6a), and the C143A/C167A/C186A (3MT) mutant of RNF208 was unable to promote polyubiquitination of Vimentin (Fig. 6b). In addition, immunoblot analysis showed that ectopic expression of the Myc-RNF208 (3MT) mutant could not induce Vimentin degradation (Fig. 6c). Therefore, C143, C167, and C186 sites within the RING domain

of RNF208 are crucial for the polyubiquitination-mediated degradation of Vimentin. Moreover, we found that RNF208 specifically bound to the head domain of Vimentin but not to the rod and tail domains (Supplementary Fig. 7c). Based on the fact that ubiquitination occurs at lysine residues of target proteins, we identified one lysine residue in the head domain of Vimentin. To examine whether this residue is critical for RNF208-mediated polyubiquitination of Vimentin, we generated a Vimentin mutant (K97A) by replacing Lys97, which is evolutionarily conserved in different species (Supplementary Fig. 7d). While wild-type Vimentin was subjected to polyubiquitination by RNF208, the Vimentin (K97A) mutant did not undergo polyubiquitination (Fig. 6d). Additionally, cycloheximide treatment revealed that the stability of the Vimentin (K97A) mutant was not influenced in the presence of RNF208 (Fig. 6e, f). Thus, the Lys97 residue of Vimentin is a major target of RNF208-induced polyubiquitination of Vimentin for regulating its stability.

Next, we investigated whether the activity of RNF208 E3 ligase directly influences the metastatic potential of TNBC cells in vivo using wild-type and the 3MT mutant RNF208-overexpressing MDA-MB-231 cells. Interestingly, RNF208 (3MT) mutant-overexpressing MDA-MB-231 cells resulted in a significant increase of cell proliferation and migration as well as metastatic pulmonary nodules in immunodeficient mice, similar to those of control cells (Fig. 6g; Supplementary Fig. 8). In accordance with this observation, immunohistochemistry staining of lung metastasis tissues showed that RNF208 overexpression decreased the expression levels of Vimentin in the tissue sections, whereas the RNF208 (3MT) mutant did not affect its expression levels (Fig. 6h). We further examined whether expression of the nondegradable K97A mutant version of Vimentin in RNF208 expression cells can rescue the metastasis phenotypes. We used siRNA targeting *VIM* to delete the endogenous Vimentin level and overexpressed wild-type Vimentin or Vimentin (K97A) mutant in control and RNF208-overexpressing MDA-MB-231

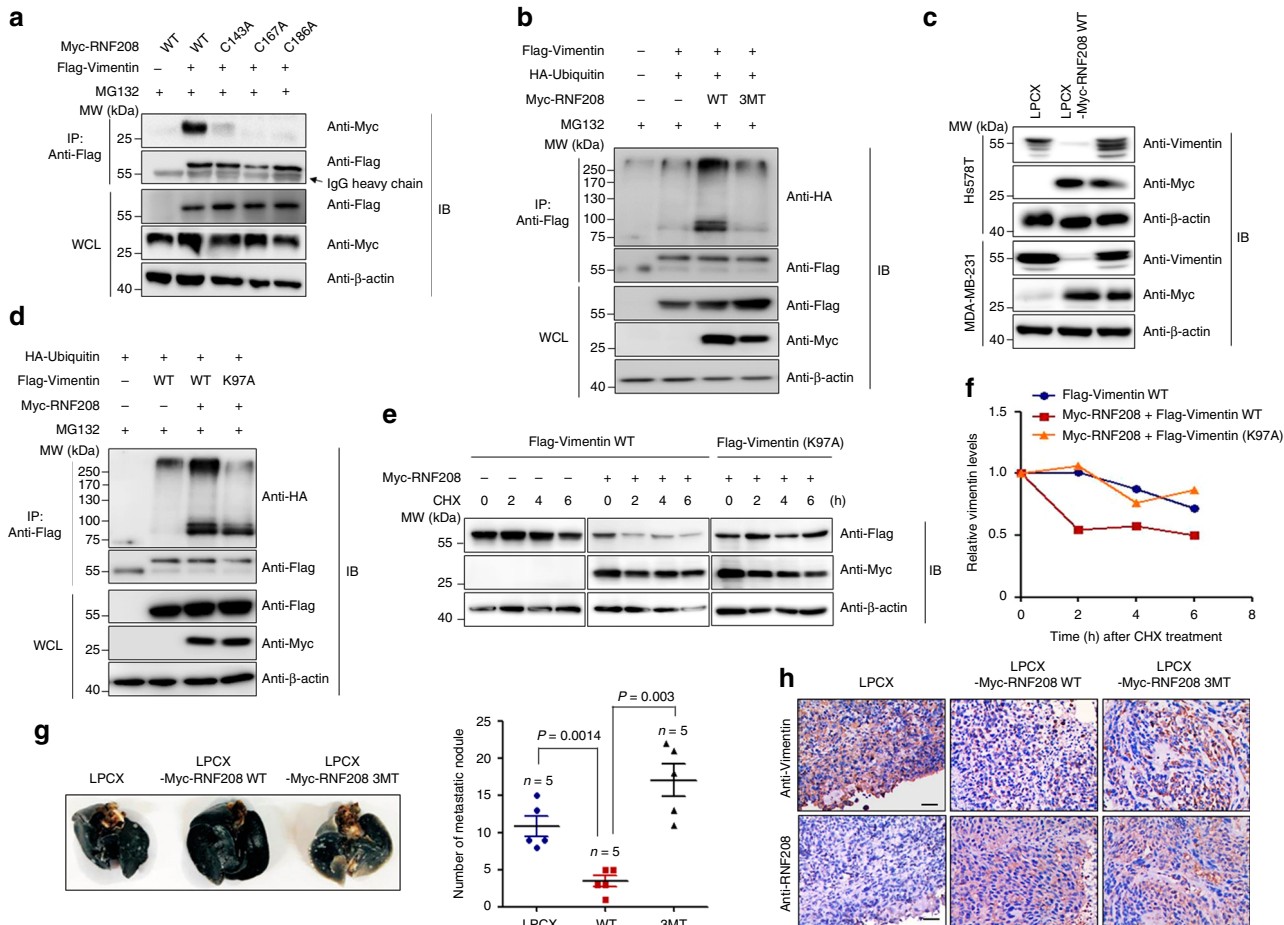

**Fig. 6 Activity of the RNF208 E3 ligase is required for the polyubiquitination-mediated degradation of Vimentin through interaction with its head domain. a** Wild-type RNF208 or RNA208 mutant (C143A, C167A, C186A) plasmids were cotransfected with Flag-Vimentin into 293T cells upon MG123 treatment. **b** After the Flag-Vimentin plasmid was cotransfected with the HA-Ubiquitin plasmid into 293T cells with wild-type RNF208 or RNF208 mutant (3MT, C143A/C167A/C186A), cell lysates were immunoprecipitated with anti-Flag antibody, and ubiquitinated Vimentin was observed by immunoblotting using an anti-HA antibody. **c** Immunoblot analysis showing the expression levels of Vimentin in control, wild-type RNF208, and RNF208 mutant (3MT)-overexpressing MDA-MB-231 cells. **d** HA-Ubiquitin plasmid was cotransfected into 293T cells with wild-type or lysine mutant (K97A) of Flag-Vimentin in the absence or presence of Myc-RNF208 plasmid. Cell lysates were immunoprecipitated with anti-Flag antibody, and ubiquitinated Vimentin was observed by immunoblotting using anti-HA antibody. **e, f** Flag-Vimentin or Flag-Vimentin (K97A) plasmids were cotransfected into 293T cells in the absence or presence of Myc-RNF208 plasmids. Cells were treated with cycloheximide (CHX, 50 μg ml$^{-1}$) for the indicated times (**e**). The data were quantified using ImageJ software (**f**), and β-actin was used as an internal control. **g** Representative whole lung image stained with India ink showing metastatic nodules from 8 weeks, derived from lateral tail-vein injection of control, wild-type RNF208 or RNF208 mutant (3MT)-overexpressing MDA-MB-231 cells (left) ($n = 5$ per group). Scatter plot showing the number of lung metastatic nodules (right). The $P$ value was calculated by unpaired two-tailed Student's $t$ tests. The data represent the mean ± SD. **h** Representative IHC image showing RNF208 and Vimentin expression in lung tissues from (**g**). Original magnification ×100. Scale bar, 50 μm. Source data for (**g**) are available in Source Data file. Unprocessed original scans of blots in (**a**−**e**) are shown in Supplementary Fig. 13.

cells. Interestingly, RNF208 overexpression markedly decreased cell migration in control cells as well as wild-type Vimentin-overexpressing cells, whereas RNF208 overexpression-induced reduction in cell migration was rescued in Vimentin (K97A) mutant-overexpressing cells (Supplementary Fig. 9a). Consistent with this observation, an increase in metastatic pulmonary nodules was observed in mice injected with RNF208/Vimentin (K97A) mutant-overexpressing cells compared with RNF208 or RNF208/wild-type Vimentin-overexpressing cells, indicating that RNF208 decreases lung metastasis by targeting the Lys97 residue of Vimentin (Supplementary Fig. 9b). Collectively, these results suggest that activity of RNF208 E3 ligase is required for the polyubiquitination-mediated degradation of Vimentin through interaction with the head domain, eventually attenuating the metastatic capacity of TNBC cells.

**RNF208 targets the phosphorylation of Vimentin at the Ser39 residue.** Studies have reported that several serine residues within the N-terminal head domain of Vimentin are phosphorylated by various kinases, and phosphorylated Vimentin in turn is disassembled and present in the soluble fraction in the cytoplasm, eventually promoting cell migration and invasiveness[13,17]. Based on the fact that phosphorylated Vimentin as a soluble form is associated with metastasis, we hypothesized that RNF208 may induce the degradation of soluble Vimentin by recognizing phosphorylated Vimentin. To assess our hypothesis, we initially confirmed the expression level of Vimentin in the soluble and insoluble fractions in RNF208-overexpressing MDA-MB-231 cells. Interestingly, RNF208 was predominantly located in the soluble fraction, and its overexpression led to a significant reduction of Vimentin in the soluble fraction compared with the insoluble

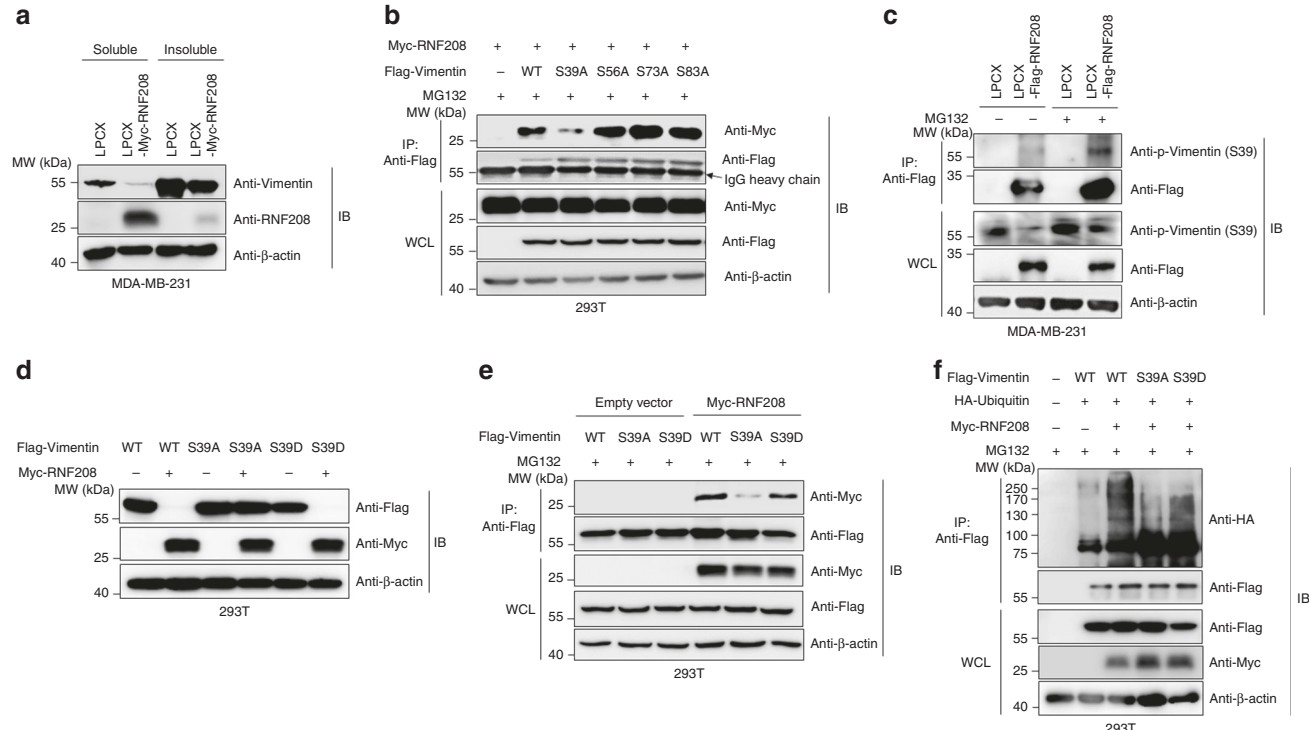

**Fig. 7 RNF208 degrades the soluble form of Vimentin by recognizing the phosphorylation of Vimentin at the Ser39 residue. a** Immunoblot analysis of the expression levels of Vimentin in soluble and insoluble fractions isolated from control and RNF208-overexpressing MDA-MB-231 cells. **b** The 293T cells were cotransfected with Myc-RNF208 and Flag-Vimentin mutant (wild-type, S39A, S56A, S73A, S83A) plasmids upon MG132 treatment. Cell lysates were immunoprecipitated with anti-Flag antibody and then immunoblotted with the indicated antibodies. **c** Immunoprecipitation assay showing the endogenous interaction between RNF208 and phosphorylated Vimentin at the Ser39 residue in RNF208-overexpressing MDA-MB-231 cells with or without MG132. Cell lysates were immunoprecipitated with anti-Flag antibody and immunoblotted with the indicated antibodies. **d** The 293T cells were cotransfected with wild-type, inactive mutant (S39A), or constitutively active mutant (S39D) of Flag-Vimentin in the absence or presence of Myc-RNF208. Cell lysates were subjected to immunoblotting with the indicated antibodies. **e** After Flag-Vimentin WT, S39A, or S39D plasmids were cotransfected into 293T cells with or without Myc-RNF208 plasmid upon MG132 treatment, cell lysates were immunoprecipitated with anti-Flag antibody and immunoblotted with the indicated antibodies. **f** HA-Ubiquitin plasmid was cotransfected into 293T cells with wild-type, S39A, or S39D plasmids of Flag-Vimentin in the absence or presence of Myc-RNF208 plasmid upon MG132 treatment. Cells lysates were immunoprecipitated with anti-Flag antibody, and ubiquitinated Vimentin was observed by immunoblotting using an anti-HA antibody. Unprocessed original scans of blots in (**a**−**f**) are shown in Supplementary Fig. 13.

fraction (Fig. 7a). Thus, we tested whether RNF208-mediated degradation of soluble Vimentin is linked to phosphorylation sites within the head domain of Vimentin, which is responsible for the interaction with RNF208. We generated various phosphorylation site mutants within the head domain of Vimentin. Interestingly, RNF208 did not bind to the Vimentin (S39A) mutant compared with wild-type protein and other serine mutants (Fig. 7b; Supplementary Fig. 10). We also investigated the physiological significance of the interaction between RNF208 and phosphorylated Vimentin at the Ser39 residue (S39) with or without MG132. RNF208 strongly interacted with endogenous phosphorylated Vimentin (S39) in the presence of MG132 compared to the control (Fig. 7c). To support this observation, we used a constitutively active mutant (S39D) of Vimentin. Ectopic expression of RNF208 did not affect the expression levels of the phosphorylation-incapable Vimentin (S39A) mutant, whereas the constitutively active Vimentin (S39D) mutant was strongly destabilized by RNF208 overexpression, similar to that of wild-type protein (Fig. 7d). Consistent with this result, an immunoprecipitation assay revealed that RNF208 interacted with wild-type and constitutively active Vimentin (S39D) upon MG123 treatment, whereas inactive Vimentin (S39A) significantly reduced the binding capability of RNF208 (Fig. 7e). We also examined whether the Ser39 residue of Vimentin was required for polyubiquitination by RNF208. Importantly, RNF208 overexpression induced polyubiquitination of wild-type as well as active Vimentin (S39D), whereas polyubiquitination of inactive Vimentin (S39A) was reduced even in the presence of RNF208 (Fig. 7f). Therefore, these results suggest that phosphorylation of Vimentin at Ser39 is critical to facilitate RNF208-mediated polyubiquitination of Vimentin and attenuation of Vimentin stability. Reportedly, phosphorylation of Vimentin at Ser39 is induced by AKT, eventually promoting cell motility and invasiveness[18]. Considering that Vimentin (S39) is increased in TNBC cells compared to luminal subtype cells (Supplementary Fig. 11a), we further examined whether RNF208 influences cell migration by targeting phosphorylation of Vimentin (S39). Indeed, ectopic expression of wild-type or active Vimentin (S39D) in MCF-7 nonmetastatic luminal cells increased cell migration compared to control or inactive Vimentin (S39A)-overexpressing cells, whereas RNF208 overexpression resulted in the reduction of cell motility in the presence of wild-type or active Vimentin (S39D) (Supplementary Fig. 11b, c). Taken together, our findings suggest that RNF208 specifically targets phosphorylated Vimentin at Ser39 as a soluble form for polyubiquitination-mediated degradation, ultimately suppressing metastasis.

## Discussion

In this study, our findings suggest a unique role of the RNF208 E3 ligase as a novel negative regulator of soluble Vimentin, thereby

inhibiting metastasis of TNBC. In addition to the overexpression of RNF208 in luminal subtype breast cancers, our results indicated that RNF208 serves as an estrogen-inducible protein that plays a role in blocking the aggressiveness of breast cancers. Furthermore, we demonstrated a mechanism by which RNF208 decreased the stability of soluble Vimentin via K27-linked polyubiquitin-mediated proteasomal degradation by recognizing phosphorylated Vimentin at Ser39, eventually attenuating TNBC progression (Supplementary Fig. 12).

Considering that ERα is a transcription factor that regulates gene expression in mammary gland development and that its dysregulation is closely associated with breast cancer progression, ERα-dependent target molecules may be associated with aggressive breast cancer progression, and their relationship remains to be elucidated. In this study, based on gene expression profiles and immunohistochemistry analyses of breast cancer specimens, we showed that RNF208 was specifically underexpressed in human TNBC cells and in primary TNBC tissues compared to luminal subtype tissues; reduced expression of RNF208 resulted in the decreased recurrence-free survival of breast cancer patients, suggesting that aberrant expression of RNF208 was strongly associated with poor clinical outcomes. Furthermore, public microarray datasets revealed a significant correlation of RNF208 expression with expression status of ERα in human breast cancers. Thus, our findings strongly suggest that RNF208 expression may be a significant prognostic marker to predict breast cancer progression, depending on ERα expression status.

Additionally, ERα expression is epigenetically silenced by DNA methyltransferase or histone deacetylases in the *ESR1* promoter of TNBC cells, leading to an aggressive phenotype and the failure of endocrine therapy[19]. Several studies have shown that re-expression of functional ERα by treatment with DNMT and HDAC inhibitors resulted in significant growth inhibition in TNBC cells[20,21]. Furthermore, although it was recently terminated, a clinical trial aimed at reactivating ERα, which can be targeted by antiestrogen agents, by a combination of DNMT and HDAC inhibitors to patients with TNBC has been conducted (ClinicalTrials.gov, NCT01194908). However, these studies raise questions about how re-expression of ERα suppresses cancer progression in TNBC cells. Considering that the lack of RNF208 in patients with TNBC seems to be due to exclusive down-regulation of ERα, it is feasible that upregulation of RNF208 induced by re-expression of ERα in TNBC cells may act as a tumor suppressor. Indeed, RNF208 is dependent on ERα re-expressed by 5-aza-dC in TNBC cells, and its expression was also upregulated through EREs within the promoter region of RNF208 upon E2 treatment, indicating that RNF208 is a novel estrogen-inducible target gene in an ERα-dependent manner. Moreover, overexpression of RNF208 markedly suppressed the tumorigenic and metastatic capacity of TNBC cells in vitro and in vivo. Therefore, we expect that the induction of RNF208 via ERα re-expression by using epigenetic agents may exert anticancer effects in ERα-negative breast cancers. Additionally, our observations may provide the mechanistic basis for the identification of patients who are most likely to benefit from ERα-targeted therapies. Furthermore, in parallel with ERα studies in hormone receptor-negative breast cancer therapy, ligand-mediated activation of ERβ has also shown potent anti-proliferative effects in TNBC cells through the suppression of metastatic phenotypes[22–24]. It is possible that RNF208 expression may be associated with the expression status of ERβ, similar to ERα. However, because the molecular classification of breast cancer subtypes is mainly defined by ERα expression and most of public gene expression profiles do not fully reflect different profiles between ERα and ERβ, we could not analyze the correlation between RNF208 and ERβ expression. Further studies will be required for deeper

insight into the role of RNF208 in ERα/β-dependent or -independent pathways during breast cancer progression. Furthermore, considering that PTMs, such as ubiquitination, sumoylation, phosphorylation, and acetylation, regulate stability of E3 ligase proteins via cross-talk with other proteins, expression of RNF208 protein may be regulated by PTMs in a context-dependent manner. Further studies will be required to better understand the PTMs of RNF208 in various cancer cells.

Although RNF208 belongs to the RING finger (RNF) protein family and serves as an E3 ubiquitin ligase, the functional roles of RNF208 have not been reported. To our knowledge, the findings presented here are the first to demonstrate an association between RNF208 and Vimentin in breast cancer progression. Our results showed that overexpression of RNF208 induced destabilization of Vimentin through K27-linked polyubiquitin-mediated proteasomal degradation, resulting in a decrease of metastasis in TNBC cells. Further supporting this observation, mutation analysis of RNF208 showed that inactivation of the RNF208 E3 ligase failed to induce degradation of Vimentin and rescued metastasis in TNBC cells. In the current study, our findings raise questions regarding how RNF208 regulates the functions of Vimentin associated with metastasis in TNBC cells. Previous studies have reported that Vimentin is functionally regulated by PTMs, especially phosphorylation, eventually contributing to enhanced tumor growth and metastasis[25]. In particular, AKT1 phosphorylates Vimentin at Ser39 residue, which leads to disassembly from the insoluble form (dephosphorylation) to the soluble form (phosphorylation), and phosphorylated Vimentin enhances cell migration and invasiveness[18]. Based on this fact, we speculated that RNF208 might induce the degradation of phosphorylated Vimentin, which promotes metastasis. Our results suggested that RNF208 specifically recognizes phosphorylated Vimentin at the Ser39 residue to facilitate polyubiquitin-mediated proteasomal degradation. Indeed, RNF208 induced the degradation of soluble Vimentin than insoluble Vimentin and specifically bound to endogenous phosphorylated Vimentin at the S39 residue. Moreover, RNF208 induced the degradation of constitutively phosphorylated Vimentin (S39D) and markedly attenuated the cell migration and invasiveness enhanced by Vimentin (S39D).

Therefore, RNF208 plays a negative role in aggressive breast cancer progression by destabilizing AKT-mediated phosphorylation of soluble Vimentin. Additionally, considering that the activity of the RNF208 E3 ubiquitin ligase may be responsible for Vimentin polyubiquitination in breast cancers, further comprehensive work is needed to understand the functional roles of RNF208 in other malignant tumors, and novel substrates ubiquitinated by RNF208 should be investigated.

In conclusion, our results present a mechanism by which RNF208, an estrogen-inducible E3 ligase, facilitates K27-linked polyubiquitin-mediated proteasomal degradation of soluble Vimentin by targeting phosphorylated Vimentin at Ser39, thereby exerting a unique function of RNF208 in breast cancer progression. In addition, given that the ERα expression status is strongly linked to breast cancer subtypes, RNF208 may be useful as a prognostic marker in breast cancers, and modulation of RNF208 in an ERα-dependent manner may be a therapeutic intervention against metastatic breast cancers.

## Methods
**Cell culture and reagents**. The human breast cancer cell lines MCF-7, T47D, Hs578T, and MDA-MB-231 were obtained from the American Type Culture Collection (ATCC); ZR-75B, cloned from the ZR-75-1 cell lines[26], was obtained from the National Cancer Institute (NCI)/NIH. The breast cancer cell lines were cultured at 37 °C in DMEM medium with 10% fetal bovine serum (FBS) and 1% penicillin/streptomycin. Medium and reagents for cell culture were purchased from WELGENE, Inc., Republic of Korea. In particular, MCF-7 and T47D cells were cultured in DMEM medium without phenol red with 10% dextran-coated charcoal

(Sigma)-stripped FBS and 1% penicillin/streptomycin for 24 h prior to 17β-estradiol treatment. The cell lines in this study were not found in the database of commonly misidentified cell lines maintained by ICLAC and NCBI Biosample and were routinely tested for mycoplasma contamination by PCR. Transfections were carried out using Fugene® HD (Promega) or Lipofectamine® RNAiMAX Reagent (Invitrogen) according to the manufacturer's instructions. Cycloheximide (01810), 17β-estradiol (E2758), 5-aza-2′-deoxycytidine (A3656), MG132 (C2211), Bafilomycin A1 (B1793), Chloroquine (C6628), 3-methyladenine (M9281) and N-ethylmaleimide (E3876) were purchased from Sigma-Aldrich.

**Plasmids.** Full-length human RNF208 complementary DNA (cDNA) was amplified from MCF-7 cDNA by PCR and subcloned into the *Eco*RI and *Xho*I sites of the pCS4-3Flag vector or pCMV-Myc vector (Addgene), resulting in Flag-RNF208 and Myc-RNF208. RNF208 cDNA was subcloned into the *Cla*I and *Not*I sites of pEBG-GST (NIH). Flag-RNF208 or Myc-RNF208 was subcloned into the *Xho*I and *Cla*I sites of the LPCX vector (NIH), resulting in LPCX-Flag-RNF208 or LPCX-Myc-RNF208, respectively. Human Flag-Vimentin was kindly provided by J.P. (AICT, Korea), and HA-Ub and HA-Ub mutants (K27/K29/K48/K63) were kindly provided by S.H.P. (Sungkyunkwan University, Korea). The human RNF208 promoter was amplified by PCR from the genomic DNA of MCF-7 cells and isolated by a HiYield™ Genomic DNA Mini Kit (Real Genomics). The amplified PCR fragment was cloned into the *Kpn*I and *Xho*I sites of the pGL3 basic vector (Promega). Primer sequences for PCR amplification in this study are listed in Supplementary Data 1.

**Generation of stable cell lines.** For generation of retroviruses, GP2-293 cells were plated on a 100-mm culture plate 24 h before transfection. Transfection was performed using polyethylenimine (PEI) with 10 μg DNA and 5 μg VSV-G per plate. For the knockdown experiment, Lenti-293 cells were seeded in 100-mm culture plates 24 h before transfection of 10 μg DNA, 10 μg Δ8.2 and 5 μg VSV-G using PEI. After transfection, the conditioned medium containing recombinant retroviruses or lentiviruses was collected and filtered through 0.45-μm sterilization filters. Then, 3 ml of filtered retroviruses was applied immediately to MDA-MB-231 and Hs578T cells, and 3 ml of filtered lentiviruses was added to T47D cells, which had been plated for 18 h before infection in a 100-mm culture dish. Polybrene (Sigma-Aldrich) was added to a final concentration of 8 μg ml$^{-1}$, and the supernatants were incubated with the cells for 8 h. The medium was aspirated and replaced with fresh viral supernatant, and the procedure was repeated. After infection, the cells were placed in fresh growth medium for 24 h and cultured as usual. Selection with 2 μg ml$^{-1}$ puromycin (Sigma-Aldrich) was initiated 48 h after infection. For transient knockdown of ERα, 20 μM of siESR1 RNA duplex (Bioneer) was transfected using Lipofectamine® RNAiMAX Reagent (Invitrogen) in target cell lines.

**RNA extraction, RT-PCR and real-time qRT-PCR.** Total RNA was isolated from cells and tissues using the easy-BLUE Total RNA extraction kit (Promega) according to the protocol provided by the manufacturer. Reverse transcription was carried out with 2 μg of purified RNA using M-MLV reverse transcriptase (Promega, M1705). The synthesized cDNA was amplified by PCR using specific primers. PCR products were visualized by electrophoresis on 1.5% agarose gels with Redsafe (Chembio, 21141) staining and analyzed with an ImageQuant LAS 4000 image analyzer (GE Healthcare). The 18S rRNA gene was used as an internal control. Quantitative real-time PCR (qRT-PCR) was performed with the proper primers using 2× SYBR Green PCR Master Mix (TaKaRa) and conducted by QuantStudio 5 (Applied Biosystems).

**RNA sequencing.** Total RNA was treated with DNase I, purified with the miR-NAeasy Mini Kit (Qiagen) and subsequently quality checked using an Agilent 2100 Bioanalyzer (Agilent). An Illumina platform (Illumina) was used to analyze transcriptomes with a 90 bp paired-end library. RNA sequencing data have been deposited in the Gene Expression Omnibus (GEO) database under accession code GSE100878.

**Chromatin immunoprecipitation.** ChIP assays using ERα ChIP grade antibody (Abcam) were performed as described previously[27]. In particular, the crosslinked cells were sonicated by a Bioruptor TOS-UCW-310-ES (output, 250W; 21 cycles of 30 s of sonication with 30 s internals; Cosmo Bio, Japan). The primer sequences are described in Supplemental Data 1. EREs on the RNF208 promoter region were predicted using PROMO v3.0.2 software. For normalization, 5% of input chromatin was used in the PCR analyses.

**CRISPR genome editing.** To delete −1131/−1124 region of RNF208 promoter, we generated guide RNA (gRNA) using GeneArt™ Precision gRNA Synthesis Kit (invitrogen) according to the protocol provided by manufacturer. MCF-7 cells were transfected with the gRNA, ssODN and Cas9 protein (Toolgen) using the Neon Transfection System following the manufacturer's protocol. To generate RNF208 knockout cells, we inserted the annealed target sequence oligo into PX459 vector (Addgene) following the Zhang lab protocol[28]. Cells were infected with lentiviral

particles upon 8 mg ml$^{-1}$ polybrene treatment. After infection, virus-containing medium was replaced with normal medium and then RNF208 knockout cells were re-selected in 2 mg ml$^{-1}$ puromycin. The target sequence and ssODN sequence informations are listed in Supplementary Data 1.

**Human breast cancer tissue microarray and immunohistochemistry.** Human breast cancer tissues from surgical section at the Gangnam Severance Hospital, Yonsei University College of Medicine (Seoul, Korea) were collected between January 1996 and December 2004 after approval by the institutional review board (IRB approval number 3-2013-0268). All procedures involving human participants were performed in compliance with the relevant ethical standards. For immunohistochemistry, each TMA slide was stained with rabbit polyclonal anti-RNF208 antibody (Abcam, ab121658) and counterstained with hematoxylin. After staining, slides were scored under a microscope and analyzed for RNF208 and Vimentin expression levels depending on the breast cancer subtype.

**In vivo tumor formation and lung metastasis.** All procedures were approved by the CHA Laboratory Animal Research Center (Seongnam, Korea) and Woojung Bio Animal facility (Suwon, Korea). For the tumor-formation assay, a total of $1 \times 10^7$ retrovirus-infected Hs578T and MDA-MB-231 cells were resuspended in 1:3 PBS/hydrogel (The Well Bioscience) solution and subcutaneously injected into 6-week-old female NOD/ShiLtJ-Rag2em1AMC Il2rgem1AMC (NRGA, Joong Ah Bio) mice ($n = 6$ per group) to measure tumor growth. Tumor size was monitored weekly starting at 4 weeks after injection, and tumor volume was calculated using the formula $V = (A \times B^2)/2$, where $V$ is volume (mm$^3$), $A$ is the long diameter (mm), and $B$ is the short diameter (mm). For the lung metastasis assay, retrovirus-infected MDA-MB-231-Luc cells ($1 \times 10^6$) were injected into female NOD/SCID mice through the tail vein. The occurrence of lung metastasis was monitored weekly starting 4 weeks after injection, and the bioluminescence signals were measured using an IVS-200 system (Xenogen Corp). Lungs were stained with India ink, and nodules were counted for quantitative analysis. Tumor tissues were fixed in 10% formalin solution and embedded in paraffin block. The slide sections were stained with the indicated antibodies for immunohistochemistry.

**Immunoprecipitation and immunoblot analysis.** Cells were washed twice in cold PBS and lysed in IP buffer (50 mM Tris, pH 7.4, 150 mM NaCl, 1% Triton X-100, 0.5% sodium deoxycholate, 2 mM EDTA and 10% glycerol) plus phosphatase and protease inhibitors (Roche). Whole-cell extracts were incubated with the appropriate primary antibodies overnight at 4 °C. Antibody-bound proteins were precipitated with Dynabeads (Thermo Fisher Scientific) or Glutathione Sepharose 4B (GE Healthcare) according to the manufacturer's protocol. The beads were washed three times with lysis buffer and then eluted in 2× SDS sample loading buffer. Eluted proteins were separated by SDS–polyacrylamide gel electrophoresis, transferred to PVDF membranes (Millipore), and detected using appropriate primary antibodies coupled with a horseradish peroxidase-conjugated secondary antibody by chemiluminescence (GE Healthcare). Rabbit antibody against RNF208 (ab175506, dilution 1:1000) and Ubiquitin (linkage-specific K27) (ab181537, dilution 1:1000) and mouse antibody against Vimentin [V9] (ab8069, dilution 1:1000), Vimentin [RV202] (ab8978, dilution 1:1000) and ERα [E115] (ab32063, dilution 1:1000) were from Abcam. Mouse antibody against Ubiquitin (sc-8017, dilution 1:1000), GST [B-14] (sc-138, dilution 1:2000) and Myc [9E10] (sc-40, dilution 1:2000) and rabbit antibody against p-Vimentin S39 (sc-16673, dilution 1:500) and HA [Y-11] (sc-805, dilution 1:1000) were from Santa Cruz. Mouse antibody against Flag [M2] (F3165, dilution 1:5000) and β-actin [AC-74] (A5316, dilution 1:10000) were from Sigma. Mouse antibody against E-cadherin (610181, dilution 1:1000) was from BD Biosciences and rabbit antibody against Cleaved Caspase-3 (#9664, dilution 1:1000) was from Cell Signaling. For fractionation of the soluble/insoluble proteins, cell lysate was lysed using RIPA buffer [20 mM Tris-HCl (pH 7.5). 150 mM NaCl, 1 mM Na$_2$EDTA. 1 mM EGTA. 1% NP-40] and dissociate the supernatant and pellet. Insoluble proteins were eluted in 2× SDS sample loading buffer after boiling for 5 min at 95 °C. The primary antibodies used are listed in Supplementary Data 2.

**Crosslinking for protein interaction analysis.** Crosslinking of RNF208-overexpressing Hs578T cells for mass spectrometry analysis was performed as described previously[29]. In particular, cells were fixed in 1% formaldehyde in PBS. Eluted proteins were loaded on SDS-polyacrylamide gel and separated by electrophoresis. Crosslinked proteins on gel were stained in Coomassie blue (Tech & Innovation) and analyzed through mass spectrometry at Korea Basic Science Institute (Daejeon, Korea). The RNF208 binding partner candidates are listed in Supplementary Data 3.

**In vitro ubiquitination assay.** Flag-Vimentin, Myc-RNF208, and HA-Ub plasmids were transfected into 293T cells, and 10 μM MG132 was added 6 h before the cell harvest. Then, 293T cells were lysed in SDS lysis buffer [10 mM Tris-HCl (pH 8.0), 150 mM NaCl, 1% SDS, 5 mM NEM, protease inhibitor]. The lysates were boiled for 10 min and were diluted tenfold with dilution buffer [10 mM Tris-HCl (pH 8.0), 150 mM NaCl, 1% Triton X-100]. The protein lysates were rotated with Flag (Sigma-Aldrich) overnight at 4 °C. Antibody-bound proteins were precipitated

with Dynabeads (Thermo Fisher Scientific). The precipitates were washed once with washing buffer A [10 mM Tris-HCl (pH 8.0), 150 mM NaCl, 1% Triton X-100, 0.1% SDS] and twice with washing buffer B [10 mM Tris-HCl (pH 8.0), 150 mM NaCl, 1% Triton X-100]. The proteins were eluted in 2× SDS sample loading buffer and were subjected to Western blot analysis.

**Luciferase assay**. RNF208 promoter luciferase was transfected into MCF-7 or T47D cells using FuGENE HD (Promega). The luciferase activities were analyzed using the Luciferase Assay System Kit (Promega) according to the manufacturer's protocol. All assays were performed in triplicate, and the luciferase activities were normalized against β-galactosidase activities.

**Cell migration and invasion assays**. Transwell migration assays were performed using Transparent PET membrane inserts (Falcon, 353097) as described in the manufacturer's protocol. A total of $5 \times 10^4$ cells were plated in the insert and incubated for 16 h. Invasion assays were performed with BioCoat Matrigel invasion chambers (Corning, 354578) as described in the manufacturer's protocol. Cells were starved in DMEM medium without FBS for 24 h. Starved cells ($1 \times 10^5$) were plated in the top chamber, which contained serum-free DMEM, and the bottom chamber contained DMEM with 10% FBS. After 24 h of incubation, noninvasive cells were removed with a cotton swab. The cells that migrated through the membrane and adhered to the lower surface of the membrane were fixed with 70% ethanol and stained with 0.05% crystal violet. The numbers of invaded cells in each field of view were quantified for statistical analysis.

**Wound-healing assay**. Cells were seeded with $1 \times 10^5$ cells in a six-well plate for 18 h in normal culture medium. The p200 tips were applied for cell scraping and wound creating. After scraping, cells were incubated in low serum medium (1% FBS contained) for the indicated times. For a comparison of the wound-healing percentage, 0 h images were taken immediately after wound creation using an inverted epifluorescence microscope (Nikon Ti-E H600L) with a ×20 objective. The wound-healing percentage was calculated at the indicated time from the images taken. The cell migration area was measured between dashed regions by ImageJ and normalized to control cells.

**Statistics and reproducibility**. Statistical significance was calculated by using GraphPad Prism 5 and SPSS version 18 software in this study. The significance of predicting relapse-free survival in patients with breast cancer was analyzed by using the log-rank Mantel−Cox test. One-way analysis of variance (ANOVA) was used to compare different breast cancer subtypes in GDC datasets. For all other comparisons, the unpaired two-tailed Student's $t$ test was used. $P < 0.05$ was considered statistically significant. All experiments were repeated at least three times. No statistical method was used to predetermine sample size. Sample size was chosen on the basis of literature in the field.

**Reporting summary**. Further information on research design is available in the Nature Research Reporting Summary linked to this article.

## Data availability

RNA sequencing data for Fig. 1a have been deposited in the NCBI GEO under accession code GSE100878. RNA sequencing datasets for Fig. 1d, e were downloaded from the Genomic Data Common (GDC) data portal (http://portal.gdc.cancer.gov) and deposited public microarray datasets for Figs. 1c, f, 2a, and Supplementary Fig. 2 are available in the GEO database under accession codes GSE2034 [30], GSE5460 [31], GSE41313 [32], and GSE68379 [33]. Relapse-free survival for Fig. 1i and Supplementary Fig. 1 was analyzed by the Kaplan−Meier Plotter analysis tool (http://kmplot.com/analysis). The source data for Figs. 1c, f, h, 2a, c, e, g, 3a, b, e, f, h, and 6g and Supplementary Figs. 3a−c, 5b, 8a, b, 9a, b and 11c have been provided as Source Data file.

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

## Acknowledgements

This work was supported by a grant from the National R&D Program for Cancer Control, Ministry for Health and Welfare, Republic of Korea (HA17C0037). K.-M.Y is

the recipient of a National Research Foundation grant of Korea (NRF-2017R1D1A1B0 3035390) funded by the Ministry of Education.

## Author contributions

K.P. and K.-M.Y. designed and conceptualized the research, did the experimental work, analyzed data, and wrote the manuscript. J.P. performed the animal experiments. S.G.A., A.O., and J.J. statistically analyzed the clinical data of breast cancer patients of Gangnam Severance Hospital. S.Y., T.U., and K.-M.Y. statistically analyzed public GDC and public datasets. S.M. designed and made RNF208 knockout cells using CRISPR system. J.L. and Y.P. provided technical assistance. K.-S.P. and S.-Y.L. participated in the study design and coordinated the study; K.-M.Y. and S.-J.K. designed and conceptualized the research, supervised the experimental work, analyzed data, and wrote the manuscript.

## Competing interests

The authors declare the following competing interests: S.-J.K. has personal financial interests as shareholders in TheragenEtex. All other authors declare no competing interests.
