## [Peer Review File · Nature Communications]

Reviewers' comments:

Reviewer #1 (Remarks to the Author):

The authors identify that RNF208 is a novel biomarker associated with triple-negative breast cancer (TNBC), an aggressive breast cancer subtype lacking effective targeted therapies, then demonstrate the positive clinical correlation between RNF208 expression and worse survival outcomes in TNBC. The authors further describe that RNF208, an estrogen-induced ubiquitin ligase, promotes the degradation of Vimentin through K27-linked ubiquitination of phosphorylated Vimentin, thereby suppressing lung metastasis of TNBC.

Overall, the question being investigated is highly significant and the findings are innovative. However, there are a number of issues in the current manuscript, which includes misconceptions in recapping previous studies, misinterpretation of experimental data, and the lack of statistical testing in several experiments. Moreover, there is no evidence demonstrating whether RNF208-inhibited cancer metastasis through vimentin ubiquitination. The significance of this study is therefore diminished. However, it should be possible to address these issues as noted below.

Specific points:

1. The authors showed that estrogen induces RNF208 expression epigenetically. The observations explains the downregulation of RNF208 in TNBC compared to ER+ breast tumors. However, this finding does not support the claim by the authors that RNF208 can potentially be used as a biomarker to identify patients who might benefit from ER α -targeted therapies. The key for TNBC patients to regain responses to ER α -targeted therapies is the reexpression of ER α . However, RNF208 downregulation does not cause ER α reexpression, but merely a consequence of lacking ER α expression in TNBC.

The authors stated in the instruction that "In breast cancer progression, the absence of ER α often causes tumorigenesis, which leads to high aggressiveness of breast cancer cells, and most ER α -negative breast cancer cells frequently metastasize to other organs, such as bone, brain, and lung, compared to ER α -positive breast cells". The statements are not quite accurate. For instance, although TNBC does not express ER α and generally has higher likelihood to metastasize compared to ER α -positive tumors, lacking ER α is not the primary etiology for the aggressiveness of TNBC. Also, despite estrogen/ER α signaling is irrelevant to TNBC, it is positively correlated with disease progression of ER α -positive subtype, which accounts for more than 60% of breast tumors. The authors should carefully go through the literature and re-assess their statements in the Introduction and Discussion sections.

2. The author showed that Ser38 is essential for the phosphorylation and degradation of Vimentin, and cited the report by Zhu et al. that phosphorylation of Vimentin at Ser38 is induced by Akt1. However, Zhu et al. showed that Akt1 promotes Vimentin phosphorylation at Ser39. In fact, there is no Ser38 present in human vimentin (it should be Ser39).

3. There are more than 20 Ser and Thr residues in the head domain of Vimentin. It's unclear why the authors chose to examine the S38 (S39?), S55, S72 and S83 sites, but not others. Since the S38A mutation did not fully abolish RNF208-mediated ubiquitination of Vimentin, it's possible that other phosphorylation sites are also involved. Additionally, the authors should elucidate whether the S38 phosphorylation of Vimentin leads to the suppression of TNBC metastasis.

4. The authors showed that RNF208 protein expression is associated with TNBC (Fig, 1g). The authors should examine Vimentin protein expression in breast cancer specimens to demonstrate the clinical correlation between RNF208 and Vimentin in vivo. A sample size of 3 for each group is too few to draw a convincing conclusion. The authors should extend their observation with a larger cohort of breast tumors.

Besides, the author should examine the expression Vimentin in Fig 3c.

5. The authors showed that RNF208 promotes K27-linked ubiquitination of Vimentin. Does the K27-linked ubiquitination of Vimentin occurs in endogenous Vimentin in TNBC cells? This should be analyzed. Does RNF208-mediated metastasis suppression depends on Vimentin ubiquitination? This should be done.

6. The authors showed that S38, C143, C168 and C186 of RNF208 are important for the binding between RNF208 and Vimentin. S38 is in the head domain and the rest residues are in the RING domain. How do these residues on RNF208 coordinate in mediating the interaction with Vimentin?

7. The authors showed that estrogen-induced RNF208 expression is attenuated by ESR1 knockdown. This is a critical notion. What is the sample size used in ESR1 siRNA group of Fig. 3a? At least three biological replicates are needed to solidify the conclusion.

8. The authors should provide statistical analyses for Fig. 2a, 2c, 2e-2h, 3a, 3b, 3e-3h and for Supplementary Fig. 3, 8 and 9c.

Reviewer #2 (Remarks to the Author):

The authors seek to understand the molecular mechanism underlying how RNF208 is regulated physiologically by ER signaling and furthermore, how RNF208 can suppress metastasis in triple negative breast cancer cells in part by promoting the ubiquitination and subsequent degradation of Vimentin. The paper is clearly written, however, the following concerns should be addressed before its publication at Nature Communications.

1. Figure 1b, the authors should explain why T47D expresses more RNF208 than ZR-75B at mRNA levels, but displays relatively reduced protein abundance. Is RNF208 regulated by post-translational modification as well?

2. Figure 2F, the input lane, should be -1131/-1124. Furthermore, as -1131/-1124 being found as the major ER binding site, it will be critical for the authors to use CRISPR to delete this ER binding site to see how it affects endogenous RNF208 expression in ER positive breast cancer cell lines.

3. Figure 3A and 3E, it will be critical for the authors to include the catalytically dead version of RNF208 (such as the 3MT mutant) as a negative control in these assays to demonstrate the critical role of E3 ligase in mediating all the phenotypes.

4. Figure 4B, it will be important to include other RNF208 interacting protein such as Annexin 1 as a negative control. It will also be critical to include the catalytically dead version of RNF208 (such as the 3MT mutant) as a negative control, as well as include the immunoblot analyses of various EMT related transcriptional factors including but not limited to Snail, Slug or Twist to ensure that RNF208 did not affect these transcriptional factors to potentially affect Vimentin mRNA levels.

5. Figure 4 in general, this figure largely relies on over physiological expression of RNF208. It will be critical for the authors to generate RNF208 knockout cell lines by CRISPR in ER positive breast cancer cell lines to examine its effects to Vimentin and migration ability.

6. Figure 5A, have the authors found cancer derived mutations in RNF208 to elevate Vimentin, will CRISPR knockin of C143A elevate Vimentin?

7. Figure 5G, the authors should examine whether expression of the non-degradable K97A mutant version of Vimentin in RNF208 expression cells can rescue the metastasis phenotypes.

8. Figure 6, the authors should compare the functional differences of WT versus S38A or S38D in dictating cell migration or metastasis.

REVIEWER #1

1. The authors showed that estrogen induces RNF208 expression epigenetically. The observations explain the downregulation of RNF208 in TNBC compared to ER+ breast tumors. However, this finding does not support the claim by the authors that RNF208 can potentially be used as a biomarker to identify patients who might benefit from ER α -targeted therapies. The key for TNBC patients to regain responses to ER α -targeted therapies is the reexpression of ER α . However, RNF208 downregulation does not cause ER α reexpression, but merely a consequence of lacking ER α expression in TNBC.

: In this study, we have shown that RNF208 is significantly underexpressed in TNBC cells compared with luminal subtypes (Main Figure 1). Agreeing with the comment from Reviewer #1, we are not claiming that RNF208 can potentially be used as a biomarker to identify patients who might benefit from ER α -targeted therapies. However, we are claiming that RNF208 is a novel ER α target gene. As mentioned in Supplementary Figure 2, the methylation levels of the RNF208 promoter were not significant in TNBC cells, indicating that methylation is not involved in RNF208 silencing. Furthermore, we showed that ER α re-expression by 5-aza-dC treatment in TNBC cells resulted in an increase in the expression level of RNF208, whereas siRNA-mediated ER α knockdown in 5-aza-dC-induced ER α re-expression in TNBC cells attenuated the RNF208 expression, implying that ER α expression is required for the induction of RNF208 (Main Figure 2i, j; Supplementary Figure S3c). Therefore, as mentioned in discussion section, considering that RNF208 downregulation seems to be due to lack of ER α , we think that upregulation of RNF208 may be induced by re-repression of ER α in TNBC cells.

The authors stated in the instruction that “In breast cancer progression, the absence of ER α often causes tumorigenesis, which leads to high aggressiveness of breast cancer cells, and most ER α -negative breast cancer cells frequently metastasize to other organs, such as bone, brain, and lung, compared to ER α -positive breast cells”. The statements are not quite accurate. For instance, although TNBC does not express ER α and generally has higher likelihood to metastasize compared to ER α -positive tumors, lacking ER α is not the primary etiology for the aggressiveness of TNBC. Also, despite estrogen/ER α signaling is irrelevant to TNBC, it is positively correlated with disease progression of ER α -positive subtype, which accounts for more than 60% of breast tumors. The authors should carefully go through the literature and re-assess their statements in the Introduction and Discussion sections.

: In response to Reviewer #1's comments, we have deleted these statements in the Introduction section and changed the sentence “Dysregulation of ER α is often associated with advanced breast cancers by altering the ER α -dependent genes that are involved in cell proliferation and metastasis” (Main text page 3, line 11-12).

2. The author showed that Ser38 is essential for the phosphorylation and degradation of Vimentin, and cited the report by Zhu et al. that phosphorylation of Vimentin at Ser38 is induced by Akt1. However, Zhu et al. showed that Akt1 promotes Vimentin phosphorylation at Ser39. In fact, there is no Ser38 present in human vimentin (it should be Ser39).

: As Reviewer #1 acknowledged in his/her comments, there are several papers determining the phosphorylation of Vimentin at Ser39 residue (Zhu et al., 2011; Yang et al., 2019) and commercial phospho-Vimentin (ser39) antibody is also sold from Cell Signaling Technology and ThermoFisher Scientific. On the other hand, several groups named phospho-Vimentin at Ser38 residue and used the endogenous antibody from Santa Cruz Biotechnology (sc-16673-R), named p-Vimentin (Ser 38)-R (Cogli et al., 2013) and from Abcam (ab52942), named anti-Vimentin (phospho S38) antibody [EP1069Y] (Kajita et al., 2014), respectively. In these papers, they didn't count on methionine as a start residue when examining the phosphorylation sites on human Vimentin (Eriksson et al.). Given that there is serine amino acid on 39th of human Vimentin protein, which includes methionine as a start residue, we have revised the term Ser38 to Ser39, Ser55 to Ser56 and Ser72 to Ser73.

Reference

Zhu, Q. S. *et al.* Vimentin is a novel AKT target mediating motility and invasion. *Oncogene* **20**, 457-470 (2011)

Yang, C. Y. *et al.* Src and SHP2 coordinately regulate the dynamics and organization of vimentin filaments during cell migration. **38**, 4075-4094 *Oncogene* (2019)

Cogli, L. *et al.* Vimentin phosphorylation and assembly are regulated by the small GTPase Rab7a. *Biochem Biophys Acta* **1833**, 1283-1293 (2013)

Kajita, M. *et al.* Filamin acts as a key regulator in epithelial defence against transformed cells. *Nat Commun* **5**, 4428 (2014).

Eriksson, J. E. *et al.* Specific in vivo phosphorylation sites determine the assembly dynamics of vimentin intermediate filaments. *J Cell Sci* **117**, 919-932 (2004).

3. There are more than 20 Ser and Thr residues in the head domain of Vimentin. It's unclear why the authors chose to examine the S38 (S39?), S55, S72 and S83 sites, but not others. Since the S38A mutation did not fully abolish RNF208-mediated ubiquitination of Vimentin, it's possible that other phosphorylation sites are also involved.

: Although 13 serine residues of Vimentin phosphorylated by several kinases were discovered, we focused on the residues as Ser39, Ser56, Ser73 and Ser83, which regulate Vimentin disassembly, according the references (Tsujiura K. *et al.*, Yamaguchi T. *et al.*, Zhu Q. S. *et al.*, Li Q. F. *et al.*, and Goto H. *et al.*). To address the Reviewer #1's concern, we tested the interaction relevance of between RNF208 and other phosphorylation sites of Vimentin. We generated various phosphorylation site mutants (S5A, S7A, S8A, S9A, S10A, S34A, S39A, S42A, S51A, S55A, S56A, S72A, S73A, S83A) within the head domain of Vimentin. Interestingly, RNF208 did not bind to the Vimentin (S39A) mutant compared with wild-type

and other serine mutants (Figure A). We have added these results to our manuscript (Supplementary Figure 10).

Figure A. RNF208 specifically interacted with Vimentin at the Ser39 residue. The 293T cells were co-transfected with Myc-RNF208 and various Flag-Vimentin mutant plasmids upon MG132 treatment. Cell lysates were immunoprecipitated with anti-Flag antibody and then immunoblotted with the indicated antibodies.

References

- Tsujimura K. *et al.* Visualization and function of vimentin phosphorylation by cdc2 kinase during mitosis. *J. Biol. Chem.* **269**, 31097-31106 (1994)
- Yamaguchi T. *et al.* Phosphorylation by Cdk1 induces Plk1-mediated vimentin phosphorylation during mitosis. *J. Cell Biol.* **171**, 431-436 (2005)
- Zhu, Q. S. *et al.* Vimentin is a novel AKT1 target mediating motility and invasion. *Oncogene* **30**, 457-470 (2011)
- Li Q.-F. *et al.* Critical role of vimentin phosphorylation at Ser-56 by p21-activated kinase in vimentin cytoskeleton signaling. *J. Biol. Chem.* **281**, 34716-34724 (2006)
- Goto H. *et al.* Aurora-B regulates the cleavage furrow-specific vimentin phosphorylation in the cytokinetic process. *J. Biol. Chem.* **278**, 8526-8530 (2003)

Additionally, the authors should elucidate whether the S38 phosphorylation of Vimentin leads to the suppression of TNBC metastasis.

: Vimentin is highly expressed and its S39 residue is phosphorylated in TNBC cells rather than luminal subtypes. In this experiment, we chose MCF-7 non-metastatic luminal cells, where Vimentin is underexpressed, to see whether the S39 phosphorylation of Vimentin regulates cell migration. Indeed, ectopic expression of wild-type or active Vimentin (S39D) increased cell migration compared to control or inactive Vimentin (S39A)-overexpressing cells, whereas RNF208 overexpression resulted in the reduction of cell motility in the

presence of wild-type or active Vimentin (S39D) (Figure B). Taken together, our findings suggest that RNF208 specifically targets phosphorylated Vimentin at Ser39 for polyubiquitination-mediated degradation, ultimately suppressing metastasis (Supplementary Fig. 11b,c).

Figure B. RNF208 overexpression decreases the cell migration by targeting the phosphorylation of Vimentin at Ser39 residue. **(a)** Phase-contrast microscopy images of an *in vitro* wound healing assay at 0 and 24 h after wounding in RNF208-overexpressing MCF-7 cells transfected with Flag-Vimentin WT, S39A, or S39D plasmids, respectively. Original magnification, 100x. **(b)** Transwell migration assays of RNF208-overexpressing MCF-7 cells transfected with Flag-Vimentin WT, S39A, or S39D plasmids, respectively. Migrated cells were counted following staining with crystal violet. *P* values were calculated by unpaired two-tailed Student's *t*-tests. The data represent means \pm S.D. of three independent experiments.

4. The authors showed that RNF208 protein expression is associated with TNBC (Fig, 1g). The authors should examine Vimentin protein expression in breast cancer specimens to demonstrate the clinical correlation between RNF208 and Vimentin in vivo. A sample size of 3 for each group is too few to draw a convincing conclusion. The authors should extend their observation with a larger cohort of breast tumors. Besides, the author should examine the expression Vimentin in Fig 3c.

: In response to Reviewer #1's comments, we examined the clinical correlation between RNF208 and Vimentin using breast cancer tissue microarrays (TMAs). Notably, the expression of RNF208 was remarkably lower in the tumor compartments of patients with TNBC compared to those with the luminal subtypes (Figure Ca,b). Furthermore, the expression levels of RNF208 and Vimentin were inversely correlated in matched tumor tissue from the patients of breast cancers (Figure Cc). In addition, expression of Vimentin was decreased in RNF208-overexpressing primary tumor tissues compared with control tissues from xenograft mouse experiment (Figure Cd). Given flow of the study, we have added this result to our manuscript (Main text page 10 (line 23-25), page 11 (line 1-11); Main Figure 5)

Figure C. RNF208 expression is inversely correlated with Vimentin expression in patients with breast cancer. **(a)** Representative IHC images showing the expression of RNF208 and Vimentin in luminal subtype or TNBC patient tumor tissues. Original magnification, 100x. Scale bar, 50 μ m. **(b)** The stacked bar charts showing the expression of RNF208 and Vimentin from TMAs of breast cancer patients. P values were calculated by Fisher exact test. Numbers in the bar charts indicate the number of patients according to IHC intensity scores. **(c)** The graph showing the RNF208 and Vimentin IHC intensity in matched tumors from TMAs of breast cancer patients. P values were calculated by paired two-tailed Student's t -tests. **(d)** Representative IHC images showing RNF208 and Vimentin expression in primary tumor tissues from **Main Figure 3b**. Original magnification, 100x. Scale bar, 50 μ m.

5. The authors showed that RNF208 promotes K27-linked ubiquitination of Vimentin. Does the K27-linked ubiquitination of Vimentin occurs in endogenous Vimentin in TNBC cells? This should be analyzed.

: We confirmed the endogenous K27-linked ubiquitination of Vimentin by RNF208, according to Reviewer #1's suggestion. As expected, an immunoprecipitation assays showed that RNF208 overexpression induced an endogenous K27-linked polyubiquitination of Vimentin in TNBC cells (Figure D). We have added these results to our manuscript (Main text page 10, line 16-18; Supplementary Figure 6c).

Figure D. RNF208 overexpression induces the endogenous K27-linked ubiquitination of Vimentin. Lysates of control or RNF208-overexpressing MDA-MB-231 cells were immunoprecipitated with anti-Vimentin antibody and then immunoblotted with the indicated antibodies.

Does RNF208-mediated metastasis suppression depends on Vimentin ubiquitination? This should be done.

: In this study, we have shown that the Lys97 residue of Vimentin is critical for RNF208-mediated its polyubiquitination. We conducted cell migration *in vitro* and lung metastasis *in vivo* for the examination whether RNF208-mediated metastasis suppression depends on Vimentin ubiquitination to address Reviewer #1's concern. We used siRNA targeting *VIM* to delete the endogenous Vimentin level and overexpressed wild-type Vimentin or Vimentin (K97A) mutant in control and RNF208-overexpressing MDA-MB-231 cells. Interestingly, RNF208 overexpression markedly decreased cell migration in control cells as well as wild-type Vimentin-overexpressing cells, whereas RNF208 overexpression-induced reduction in cell migration was rescued in Vimentin (K97A) mutant-overexpressing cells (Figure Ea). Consistent with this observation, an increase in metastatic pulmonary nodules was observed in mice injected with RNF208/Vimentin (K97A) mutant-overexpressing cells compared with RNF208 or RNF208/wild-type Vimentin-overexpressing cells, indicating that RNF208 decreases lung metastasis by targeting the Lys97 residue of Vimentin (Figure Eb). We have added these results to our manuscript (Main text page 13, line 3-13; Supplementary Figure 9).

Figure E. RNF208 suppresses metastasis by targeting the Lys97 residue of Vimentin. **(a)** Transwell migration assays by RNF208 overexpression in Vimentin-rescued MDA-MB-231 cells. Cells were transfected with specific *VIM* siRNA, followed by transfection with GFP-Vimentin WT or K97A mutant plasmids, respectively. Migrated cells were counted following staining with crystal violet. *P* values were calculated by unpaired two-tailed Student's *t*-tests and error bars indicate the mean \pm S.D. of three independent experiments. **(b)** Representative whole lung image stained with India ink showing metastatic nodules from 8 weeks, derived from lateral tail-vein injection of control, RNF208 WT, RNF208 WT/Vimentin WT or RNF208 WT/Vimentin (K97A) mutant-overexpressing MDA-MB-231 cells (upper). Scatter plot showing the number of lung metastatic nodules (bottom). *P* value was calculated by unpaired two-tailed Student's *t*-tests. The data represent the mean \pm S.D.

6. The authors showed that S38, C143, C168 and C186 of RNF208 are important for the binding between RNF208 and Vimentin. S38 is in the head domain and the rest residues are in the RING domain. How do these residues on RNF208 coordinate in mediating the interaction with Vimentin?

: As mentioned in this study, S39 residue is included in the head domain of Vimentin, whereas the RING domain of RNF208 possessed C143, C167, and C186 residues, which harbors the E3 ligase activity of RNF208. We showed that the Lys97 residue of the head domain of Vimentin is a major target of RNF208-induced polyubiquitination of Vimentin for regulating its stability (Main Figure 6; Supplementary Figure 7). Furthermore, phosphorylation of Vimentin at Ser39 is critical to facilitate RNF208-mediated polyubiquitination of Vimentin (Main Figure 7). Therefore, these results suggest that RNF208 promotes polyubiquitination-mediated Vimentin degradation by recognizing phosphorylated Vimentin at Ser39 (Supplementary Figure 12b).

7. The authors showed that estrogen-induced RNF208 expression is attenuated by ESR1 knockdown. This is a critical notion. What is the sample size used in ESR1 siRNA group of Fig. 3a? At least three biological replicates are needed to solidify the conclusion.

: Because RNF208 expression was increased in luminal breast cancer subtypes, which are ER α -positive, we investigated the relationship between RNF208 and ER α expression using public microarray datasets (GSE2034; GSE5460). The expression of RNF208 was positively correlated with the ER α status in breast cancers (Main Figure 2a). We have added the sample size in ER α -positive or negative breast cancer patients (Main Figure 2a).

As shown in Main Figure 2c, we demonstrated that siRNA-induced *ESR1* knockdown attenuated the E2-induced expression level of *RNF208* in T47D cells, indicating that RNF208 expression may depend on ER α expression in luminal breast cancer cells. This data was represented the mean \pm S.D. of three independent experiments. We have added in Main Figure 2 of Figure legend sections.

As shown in Main Figure 3a, we demonstrated that RNF208 overexpression markedly reduced the proliferation of TNBC cells. Sample size is mentioned in Main Figure 3 of Figure legend sections.

8. The authors should provide statistical analyses for Fig. 2a, 2c, 2e-2h, 3a, 3b, 3e-3h and for Supplementary Fig. 3, 8 and 9c.

: We have added the statistical analyses according to Reviewer #1's suggestion.

Reviewer #2 (Remarks to the Author):

The authors seek to understand the molecular mechanism underlying how RNF208 is regulated physiologically by ER signaling and furthermore, how RNF208 can suppress metastasis in triple negative breast cancer cells in part by promoting the ubiquitination and subsequent degradation of Vimentin. The paper is clearly written, however, the following concerns should be addressed before its publication at Nature Communications.

1. Figure 1b, the authors should explain why T47D expresses more RNF208 than ZR-75B at mRNA levels, but displays relatively reduced protein abundance. Is RNF208 regulated by post-translational modification as well?

: Post-translational modifications (PTMs), such as ubiquitination, sumoylation, phosphorylation, and acetylation, regulate stability of E3 ligase proteins both independently and via cross-talk with other proteins. Therefore, agreeing with the comment from Reviewer #2, we think that expression of RNF208 protein may be regulated by PTMs in a context-dependent manner. Further studies will be required to better understand the PTM of RNF208 in various cancer cells. We have explained Reviewer #2's comment in discussion section (Main text page 17, line 14-18)

2. Figure 2F, the input lane, should be -1131/-1124. Furthermore, as -1131/-1124 being found as the major ER binding site, it will be critical for the authors to use CRISPR to delete this ER binding site to see how it affects endogenous RNF208 expression in ER positive breast cancer cell lines.

: We corrected the name “-1,131/-1,126” to “-1,131/-1,124 in the input lane of Main Figure 2f. As shown in Figure Aa, we deleted position -1,131 to -1,124 bp of RNF208 promoter, which contains the Estrogen Responsive Element (ERE) sequence, using CRISPR knockout system to see how major ER α binding site affects endogenous RNF208 expression in ER α positive breast cancer cell lines according to Reviewer #2's suggestion (Figure Aa). Interestingly, E2 treatment increased expression levels of RNF208 mRNA and protein, whereas knockout of position -1,131 to -1,124 bp was not responsive, suggesting that its region of RNF208 promoter is a critical motif for ER α response (Figure Ab). We have added these new results to our manuscript (Main Figure 2h; Main text page 7).

Figure A. RNF208 expression is transcriptionally activated by ER α in luminal breast cancer cells. **(a)** Illustration of knockout of ER α -binding sites in the RNF208 promoter sequences. **(b)** RT-PCR and immunoblot analysis showing RNF208 expression upon E2 treatment for 24 h or 48 h, respectively, in control or ERE knockout MCF-7 ER α -positive breast cancer cells.

3. Figure 3A and 3E, it will be critical for the authors to include the catalytically dead version of RN208 (such as the 3MT mutant) as a negative control in these assays to demonstrate the critical role of E3 ligase in mediating all the phenotypes.

: We conducted the experiment for the examination of cell growth and migration in RNF208 (3MT) mutant-overexpressing cells to address Reviewer #2's concern. RNF208 (3MT) mutant resulted in a significant increase of cell growth and migration, compared to wild-type RNF208-overexpressing TNBC cells (Figure B). Given flow of the study, we have added these new results to our manuscript (Supplementary Figure 8; Main text page 12, line 20-23).

Figure B. Activity of RNF208 E3 ligase regulates cell proliferation and migration of TNBC cells. **(a)** Cell doublings of TNBC cells stably expressing wild-type RNF208 or RNF208 (3MT) mutant proteins. Each point represents the mean of cell numbers counted in triplicate dishes. **(b)** Transwell migration assays of wild-type RNF208 or RNF208 (3MT) mutant-overexpressing TNBC cells. Migrated cells were counted following staining with crystal violet. All *P* values were calculated by unpaired two-tailed Student's *t*-tests. These data represent means \pm S.D. of three independent experiments.

4. Figure 4B, it will be important to include other RNF208 interacting protein such as Annexin 1 as a negative control. It will also be critical to include the catalytically dead version of RNF208 (such as the 3MT mutant) as a negative control, as well as include the immunoblot analyses of various EMT related transcriptional factors including but not limited to Snail, Slug or Twist to ensure that RNF208 did not affect these transcriptional factors to potentially affect Vimentin mRNA levels.

: We examined whether RNF208 mediates degradation of Annexin A1, which is identified as RNF208 binding partner in our study, similar to Vimentin according to Reviewer #2's suggestion. Although RNF208 interacted with Annexin A1, RNF208 overexpression did not influence the expression level of its protein, suggesting that RNF208 may specifically regulate the stability of Vimentin protein (Figure C). Given flow of the study, we did not include these results in our manuscript (Data not shown).

Figure C. RNF208 did not regulate the stability of Annexin A1 protein. (a) Immunoprecipitation assay showing the endogenous interaction between RNF208 and Annexin A1. Cell lysates were immunoprecipitated with anti-Myc antibody and then immunoblotted with the indicated antibodies. (b) Immunoblot analysis showing the expression level of Annexin A1 protein in control and RNF208-overexpressing TNBC cells. Cell lysates were subjected to immunoblotting with the indicated antibodies.

In response to Reviewer #1's comments, we tested whether RNF208 (3MT) mutant influences the stability of Vimentin protein. Overexpression of RNF208 (3MT) mutant could not induce Vimentin degradation, compared to wild-type RNF208 in TNBC cells (Figure D). We have changed these results to our manuscript (Main Figure 6c).

Figure D. Activity of the RNF208 E3 ligase is required for the polyubiquitination-mediated degradation of Vimentin through interaction with its head domain. Immunoblot analysis showing the expression levels of Vimentin in control, wild-type RNF208, and RNF208 (3MT) mutant-overexpressing TNBC cells. Cell lysates were subjected to immunoblotting with the indicated antibodies.

Furthermore, we examined whether RNF208 regulates expression of EMT-related transcriptional factors to address the Reviewer #1's concern. RNF208 overexpression did not influence expression levels of ZEB1 mRNA and protein. Expression of Snail mRNA and protein was decreased in RNF208-overexpressing Hs578T cells compared with control cells, whereas RNF208-overexpressing MDA-MB-231 cells downregulated expression of Snail protein without affecting its mRNA expression. In case of Slug, its protein expression was decreased in RNF208-overexpressing Hs578T cells only. Furthermore, expression of Twist mRNA was increased in RNF208-overexpressing Hs578T cells only without affecting protein expression (Figure E). Based on these results, we assume that RNF208 may regulate expression of EMT master genes in a context-dependent manner. However, considering that RNF208 regulates the stability of Vimentin without affecting mRNA expression in both Hs578T and MDA-MB-231 cells, we think that Vimentin protein may be a specific target of RNF208 in TNBC cells. Given the complex interpretation of these results, we did not include these results in our manuscript (Data not shown).

Figure E. Expression levels of EMT master genes in RNF208-overexpressing TNBC cells. RT-PCR and immunoblot analysis showing expression levels of EMT-related transcription factors in control or RNF208-overexpressing TNBC cells.

5. *Figure 4 in general, this figure largely relies on over physiological expression of RNF208. It will be critical for the authors to generate RNF208 knockout cell lines by CRISPR in ER positive breast cancer cell lines to examine its effects to Vimentin and migration ability.*

: We could not confirm RNF208-regulated expression of Vimentin in RNF208 knockout ER α positive breast cancer cells because Vimentin is underexpressed. In this study, we have shown that knockdown of RNF208 did not change the expression of E-cadherin and Vimentin or the cell migration ability of T47D ER α positive breast cancer cell lines (Supplementary Figure 5). Consistent with this observation, RNF208 knockout by CRISPR did not influence expression of Vimentin and cell migration in T47D ER α -positive breast cancer cell lines (Figure F). We have added these results to our manuscript (Supplementary Figure 5).

Figure F. Loss of RNF208 does not influence aggressive phenotype in ER α positive breast cancer cells. (a) Immunoblot analysis showing E-cadherin and Vimentin expression in RNF208 knockdown or knockout T47D cells. Cell lysates were immunoblotted with the

indicated antibodies. **(b)** Transwell migration assays of RNF208 knockdown or knockout T47D cells. Migrated cells were counted following staining with crystal violet. The data represent means \pm S.D. of three independent experiments.

6. Figure 5A, have the authors found cancer derived mutations in RNF208 to elevate Vimentin, will CRISPR knockin of C143A elevate Vimentin?

: In response to Reviewer #1’s comments, we investigated cancer-associated RNF208 mutation using cBioportal database (<http://www.cbioportal.org>). Although RNF208 mutation was revealed 23 cases of missense mutations and 7 cases of truncating mutations among 10,437 samples of TCGA PanCancer Altas, its mutations were not reported the correlation of cancer-derived mutation in OncoKB driver annotation yet, and C143, C167, C186 residues of RNF208 were not mutated.

Furthermore, Reviewer #2’s suggested whether CRISPR knockin of RNF208 (C143A) mutant increases expression of Vimentin. We tried many times to make RNF208 (C143A) mutant knockin using CRISPR system in MDA-MB-231 cells. However, we failed the manufacture of RNF208 (C143A) mutant knockin cells because of non-specific deletion or insertion in target site (Figure G). Instead, given that RNF208 is significantly underexpressed in TNBC cells, we think that experiment of CRISPR knockin of RNF208 C143A in TNBC cells is not feasible because even though we successfully generated endogenous RNF208 C143A knockin, this gene cannot be transcribed. Therefore, we generated TNBC cells stably overexpressing RNF208 (3MT) mutant protein. Ectopic expression of the Myc-RNF208 (3MT) mutant could not induce Vimentin degradation (Main Figure 6C). In conclusion, we believe that the catalytically inactive RNF208 mutant does not affect degradation of Vimentin protein.

Figure G. Illustration of C143A knockin using CRISPR system in RNF208 transcription region of MDA-MB-231 TNBC cells.

7. Figure 5G, the authors should examine whether expression of the non-degradable K97A mutant version of Vimentin in RNF208 expression cells can rescue the metastasis phenotypes.

: We examined cell migration *in vitro* and lung metastasis *in vivo* whether expression of the non-degradable K97A mutant version of Vimentin in RNF208 expression cells can rescue the metastasis phenotypes according to Reviewer #2's suggestion. We used siRNA targeting *VIM* to delete the endogenous Vimentin level and overexpressed wild-type Vimentin or Vimentin (K97A) mutant in control and RNF208-overexpressing MDA-MB-231 cells. Interestingly, RNF208 overexpression markedly decreased cell migration in control cells as well as wild-type Vimentin-overexpressing cells, whereas RNF208 overexpression-induced reduction in cell migration was rescued in Vimentin (K97A) mutant-overexpressing cells (Figure Ha). Consistent with this observation, an increase in metastatic pulmonary nodules was observed in mice injected with RNF208/Vimentin (K97A) mutant-overexpressing cells compared with RNF208 or RNF208/wild-type Vimentin-overexpressing cells, indicating that RNF208 decreases lung metastasis by targeting the Lys97 residue of Vimentin (Figure Hb). We have added these results to our manuscript (Main text page 13, line 3-13; Supplementary Figure 9).

RNF208 WT/Vimentin (K97A) mutant-overexpressing MDA-MB-231 cells (upper). Scatter plot showing the number of lung metastatic nodules (bottom). *P* value was calculated by unpaired two-tailed Student's *t*-tests. The data represent the mean \pm S.D.

8. Figure 6, the authors should compare the functional differences of WT versus S38A or S38D in dictating cell migration or metastasis.

: In revised manuscript, we counted methionine as a start residue and corrected the term S38 to S39. In response to Reviewer #1's comments, we confirmed the functional differences of WT versus S39A or S39D in dictating cell migration. Indeed, ectopic expression of wild-type or active Vimentin (S39D) in MCF-7 non-metastatic luminal cells increased cell migration compared to control or inactive Vimentin (S39A)-overexpressing cells, whereas RNF208 overexpression resulted in the reduction of cell motility in the presence of wild-type or active Vimentin (S39D) (Figure I). Taken together, our findings suggest that RNF208 specifically targets phosphorylated Vimentin at Ser39 as a soluble form for polyubiquitination-mediated degradation, ultimately suppressing metastasis (Supplementary Fig. 11b,c).

Figure I. RNF208 overexpression decreases the cell migration by targeting the phosphorylation of Vimentin at Ser39 residue. **(a)** Phase-contrast microscopy images of an *in vitro* wound healing assay at 0 and 24 h after wounding in RNF208-overexpressed MCF-7 cells transfected with Flag-Vimentin WT, S39A, or S39D plasmids, respectively. Original magnification, 100x. **(b)** Transwell migration assays of RNF208-overexpressing MCF-7 cells transfected with Flag-Vimentin WT, S39A, or S39D plasmids, respectively. Migrated cells were counted following staining with crystal violet. *P* values were calculated by unpaired two-tailed Student's *t*-tests. The data represent means \pm S.D. of three independent experiments.

REVIEWERS' COMMENTS:

Reviewer #1 (Remarks to the Author):

The authors have addressed my previous concerns.

Reviewer #2 (Remarks to the Author):

The authors have addressed most of the raised concerns during this round of revision.